# Downregulation of ribosome biogenesis during early forebrain development

Kevin F Chau[1,2], Morgan L Shannon[1], Ryann M Fame[1], Erin Fonseca[1], Hillary Mullan[1], Matthew B Johnson[3], Anoop K Sendamarai[1], Mark W Springel[1], Benoit Laurent[4,5], Maria K Lehtinen[1,2]*

[1]Department of Pathology, Boston Children's Hospital, Boston, United States; [2]Program in Biological and Biomedical Sciences, Harvard Medical School, Boston, United States; [3]Division of Genetics, Boston Children's Hospital, Boston, United States; [4]Division of Newborn Medicine and Epigenetics Program, Department of Medicine, Boston Children's Hospital, Boston, United States; [5]Department of Cell Biology, Harvard Medical School, Boston, United States

**Abstract** Forebrain precursor cells are dynamic during early brain development, yet the underlying molecular changes remain elusive. We observed major differences in transcriptional signatures of precursor cells from mouse forebrain at embryonic days E8.5 vs. E10.5 (before vs. after neural tube closure). Genes encoding protein biosynthetic machinery were strongly downregulated at E10.5. This was matched by decreases in ribosome biogenesis and protein synthesis, together with age-related changes in proteomic content of the adjacent fluids. Notably, c-MYC expression and mTOR pathway signaling were also decreased at E10.5, providing potential drivers for the effects on ribosome biogenesis and protein synthesis. Interference with c-MYC at E8.5 prematurely decreased ribosome biogenesis, while persistent c-MYC expression in cortical progenitors increased transcription of protein biosynthetic machinery and enhanced ribosome biogenesis, as well as enhanced progenitor proliferation leading to subsequent macrocephaly. These findings indicate large, coordinated changes in molecular machinery of forebrain precursors during early brain development.

DOI: https://doi.org/10.7554/eLife.36998.001

*For correspondence:
maria.lehtinen@childrens.harvard.edu

**Competing interests:** The authors declare that no competing interests exist.

## Introduction

Neural tube closure (neurulation) is a fundamental milestone of early brain development, yet relatively little is known about the cellular and molecular transitions occurring in neural precursor cells before and after this process due to experimental challenges inherent to investigating this nascent organ (*Greene and Copp, 2014*; *Massarwa and Niswander, 2013*; *Wallingford et al., 2013*; *Wilde et al., 2014*). Prior to neural tube closure, the neural plate is home to multipotent neural stem cells, including forebrain neurectodermal precursor cells. After neural tube closure, these neurectodermal precursors become progressively lineage restricted as neuroepithelial cells, and then radial glial cells, ultimately giving rise to all neurons and glia in the adult forebrain (*Bjornsson et al., 2015*). As these progenitors proliferate, their spatial patterning serves as a blueprint for the maturing brain (*Rallu et al., 2002*; *Sur and Rubenstein, 2005*). While genes involved in driving the more mature stages of forebrain development are becoming better understood, remarkably little is known about the key genes orchestrating the function of earlier neurectodermal precursors.

While transcriptional regulation is essential for the specification and maturation of the early forebrain, less is known about the dynamics of protein biosynthesis at this early stage. Recent studies have begun to explore how regulated protein synthesis is critical for the successful construction and function of healthy cells and organs (*Fujii et al., 2017*; *Kondrashov et al., 2011*; *Pilaz et al., 2016*;

*Roko Rasin and Silver, 2016*; *Shi and Barna, 2015*). In turn, the regulation of protein biosynthetic machinery has emerged as a tunable program that can instruct cellular transitions between stem cell dormancy, proliferation, and differentiation (*DeBoer et al., 2013*; *Fujii et al., 2017*; *Khajuria et al., 2018*; *Kraushar et al., 2016*; *Sanchez et al., 2016*; *Scognamiglio et al., 2016*). Mutations in genes encoding ribosomal proteins are associated with neural tube closure defects (NTD; *Greene and Copp, 2014*; *Wilde et al., 2014*), suggesting that regulation of protein biosynthesis is critical during the earliest stages of forebrain development as well. Proteomic analyses have also revealed that ribosomal and translational proteins are elevated in amniotic fluid (AF) prior to neurulation, and are substantially decreased in nascent cerebrospinal fluid (CSF) following neurulation (*Chau et al., 2015*). However, the mechanisms leading to these changes in the AF and CSF proteomes remain incompletely understood, as this developmental stage precedes choroid plexus development and its secretion of factors into the CSF (*Hunter and Dymecki, 2007*; *Lehtinen et al., 2011*; *Lun et al., 2015*).

Here, we used RNA sequencing to reveal the transcriptomic signature of presumptive forebrain precursor cells before and after neurulation. High expression of the protein biosynthetic machinery together with elevated protein synthesis emerged as a signature of early neural precursors. These transcriptional and cell biological changes closely mirrored proteomic changes in the adjacent AF and CSF. Many genes that were downregulated after neurulation are known, direct targets of the transcription factor c-MYC (hereafter MYC) in other cell types (*Ben-Porath et al., 2008*; *Zeller et al., 2003*). Accordingly, MYC modulated ribosome biogenesis in forebrain precursors. Its forced, persistent expression in neural progenitors by mouse genetics approaches increased transcription of protein biosynthetic machinery and was accompanied by increased proliferation of radial glial progenitors leading to macrocephaly by birth. Taken together, our data identify regulation of protein biosynthetic machinery as an important signature of early forebrain development.

## Results

### Transcriptome signature of early forebrain neuroepithelium

To define the identity and biology of developing forebrain neuroepithelial cells, we microdissected the neuroepithelium away from the adjacent mesenchyme and surface ectoderm in E8.5 and E10.5 embryos (*Figure 1A*, *Chau et al., 2015*), and performed next-generation RNA sequencing (RNAseq) analysis (*Figure 1*). Gene expression analysis identified 3898 genes (q < 0.05) with significantly different expression patterns between the two ages, with 2375 genes enriched in E8.5 neuroepithelium, and 1523 genes enriched in E10.5 neuroepithelium (*Figure 1B*, *Figure 1—figure supplement 1A*).

Among the differentially expressed genes, many were secreted factors and receptors involved in signaling pathways with cardinal roles in brain development including WNT and BMP/TGFβ (*Figure 1C,D*, *Supplementary file 1*; *Monuki, 2007*; *Sur and Rubenstein, 2005*; *Wilde et al., 2014*). Some secreted factors (e.g. BMP1 and SHH) were enriched in both E10.5 progenitors and CSF, suggesting their secretion into the adjacent fluid (*Supplementary file 1*; *Chau et al., 2015*), while factors known to be involved in organismal development and neural tube closure including *Wnt5a* and *Pax3* were enriched in E8.5 (*Supplementary file 1*). Differential gene expression was further validated by quantitative RT-PCR (qRT-PCR) on 81 genes including transcription factors, cell surface receptors, and secreted factors, many of which showed an overall positive correlation (*Figure 1—figure supplement 1B*). Expression of *Glast* and *Blbp* were enriched in E10.5 progenitors, indicating the transition from neuroepithelial cells to radial glial cells (*Figure 1—figure supplement 1C*).

We next determined the biological functions of the most differentially expressed genes at each age. Consistent with the progressive lineage restriction of progenitors, initiation of neurogenesis, and patterning of the brain, the most enriched gene category at E10.5 was related to neuronal differentiation (e.g. *Ngn1*, *Blbp*, *Glast*, *Tbr2*, *Bmp4*; *Figure 1F*, *Supplementary file 1*). However, unexpectedly, the three most enriched gene categories in E8.5 neuroepithelium were related to protein biosynthetic machinery (*Figure 1E*, *Supplementary file 1*) and included genes encoding ribosomal proteins (e.g. *Rpl24*), genes involved in ribosome biogenesis (e.g. *Fbl*, *Dkc1*), and translation factors (e.g. *Eif4e*). Gene set enrichment analysis (GSEA; *Subramanian et al., 2005*) further confirmed that genes involved in ribosome biogenesis and protein synthesis were significantly enriched in E8.5

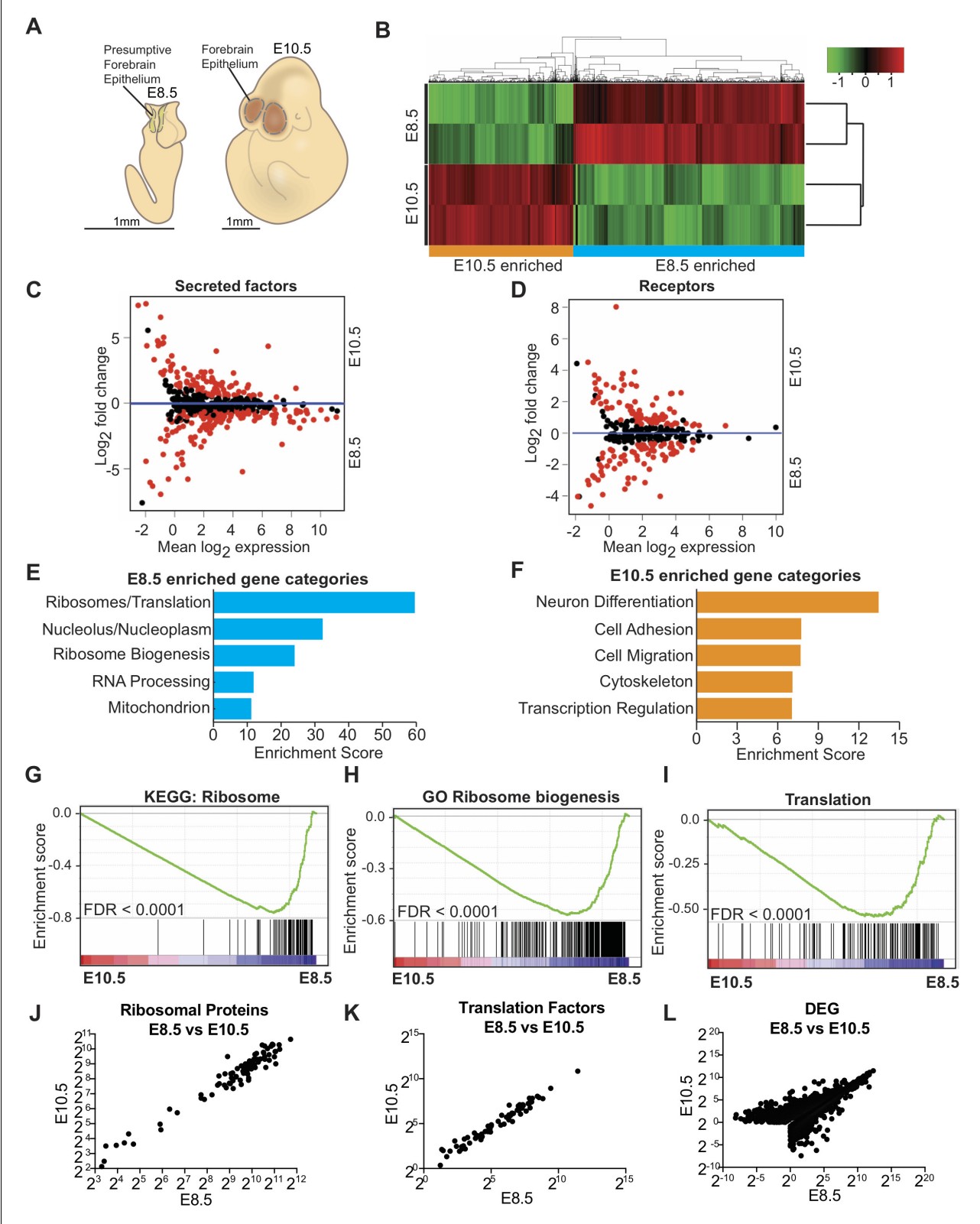

**Figure 1.** Transcriptome analysis of microdissected forebrain epithelium reveals downregulation of genes encoding protein biosynthetic machinery. (**A**) Schematic of E8.5 embryo with open forebrain neural tube (left) and E10.5 embryo (right). Shaded regions encircled by dotted line denote developing forebrain epithelium microdissected for RNAseq. (**B**) Heatmap and hierarchical clustering of ~3900 differentially expressed genes (q < 0.05): 2375 genes were enriched in E8.5 and 1523 genes were enriched in E10.5. Each biological replicate contained tissue pooled from one litter of embryos. Red and

*Figure 1 continued on next page*

*Figure 1 continued*

green indicate relatively higher and lower expression, with gene FPKM values log2 transformed, centered and scaled by rows for display purposes. (**C, D**) MA plot displaying genes encoding secreted factors (**C**), and receptors (**D**). Each dot represents a single gene. Red dots denote differentially expressed genes as identified by Cuffdiff (q < 0.05). Genes below blue line (y = 0) are enriched in E8.5. (**E**) Functional annotation clustering of E8.5 neuroepithelium enriched genes revealed overrepresentation of genes encoding ribosomal proteins, ribosome biogenesis and translation factors. The top five enriched functional clusters are shown. (**F**) Functional annotation clustering of E10.5 neuroepithelium enriched genes shows overrepresentation of genes needed for neuron differentiation. The top five enriched functional clusters are shown. (**G–I**) GSEA of E8.5 versus E10.5 neuroepithelium for gene sets involved in ribosome biogenesis and translation. Broad Institute Molecular Signatures Database Identifiers: KEGG_RIBOSOME (**G**), GO_RIBOSOME_BIOGENESIS (**H**), and TRANSLATION (**I**). Each line represents a single gene in the gene set. Genes on the right side are enriched in E8.5. (**J–L**) Correlation plots of average expression (log2 transformed FPKM) at E8.5 and E10.5 for ribosomal proteins (**J**), translation factors (**K**), and all differentially expressed genes (**L**). In all cases correlation was significant; ribosomal proteins (**J**), Spearman R = 0.91, p<0.0001; translation factors (**K**) Spearman R = 0.98, p<0.0001; and DEG (**L**), Spearman R = 0.82, p<0.0001.

DOI: https://doi.org/10.7554/eLife.36998.002

The following figure supplement is available for figure 1:

**Figure supplement 1.** Differential gene expression between E8.5 and E10.5 neuroepithelium.

DOI: https://doi.org/10.7554/eLife.36998.003

progenitors (*Figure 1G–I*). MA plots (expression ratio [M] vs. average intensity [A], log transformed) provided an overview of the expression changes of individual genes, revealing that the majority of genes encoding ribosomal proteins (*Figure 2A*), ribosome biogenesis (*Figure 2B*), and translation factors (*Figure 3A*), were enriched in E8.5 neuroepithelium. Expression of ribosomal protein or translation factor genes at E10.5 vs. E8.5 showed a positive correlation (R = 0.91 and 0.98 respectively; *Figure 1J,K*), indicating that despite downregulation of most ribosomal and translation factor genes, their stoichiometry remained similar at the two ages. Expression levels of differentially expressed genes at E10.5 vs. E8.5 also showed a positive correlation (R = 0.82, *Figure 1L*). There was no correlation between the average expression levels of genes and their fold changes between the two ages (R = −0.1696). Collectively, our data provide transcriptomic signatures of developing forebrain precursors and uncover an overall downregulation of genes encoding protein biosynthetic machinery during the inception of the mammalian forebrain.

## Decreased ribosome biogenesis and protein synthesis in E10.5 neuroepithelium

The higher expression of genes associated with ribosomes, ribosome biogenesis, and protein translation in early E8.5 precursors compared to more committed forebrain progenitors at E10.5 suggested that the protein biosynthetic machinery may be differentially regulated during early forebrain development. Ribosomal RNA (rRNA) transcription and initial assembly of pre-ribosomes occurs in nucleoli. As increased ribosome biogenesis is associated with larger nucleoli (*Silvera et al., 2010*), nucleolar volume provides a proxy for ribosome biogenesis (*Baker, 2013*; *Sanchez et al., 2016*). We visualized nucleoli with Fibrillarin (*Figure 2C*), acquired z-stack images of the developing neural tissue, and performed 3D-reconstructions of individual nucleoli in neural precursors (*Figure 2D*). Quantification of nucleolar volume revealed that E8.5 forebrain precursors had larger nucleoli compared to more mature forebrain progenitors at E10.5 (*Figure 2E*). No further reduction in nucleolar volume was observed between E10.5 neuroepithelial cells and E14.5 radial glial progenitors of the cerebral cortex (*Figure 2E*), suggesting that the E8.5 to E10.5 transition represents an important regulatory stage for ribosome biogenesis in the early forebrain.

Focusing on E8.5 and E10.5 neuroepithelia, we observed higher 5.8S pre-rRNA levels in E8.5 vs. E10.5 progenitors by fluorescence in situ hybridization (FISH; *Figure 2F*), and a modest decrease in 5.8S total rRNA at E10.5 (*Figure 2G*) that was supported by Y10b immunostaining (*Figure 2H*). Quantification of 5.8S pre-rRNA signal showed larger nucleolar area in E8.5 progenitors (*Figure 2I, J*), consistent with the fibrillarin quantification (*Figure 2E*). In agreement with these findings, ribosomal proteins including RPL11 and RPS12, which have important roles in the assembly of ribosomal subunits, were more highly expressed at E8.5 vs. E10.5 (*Figure 2K,L*; also *Chau et al., 2015*). On the other hand, expression of RPL10A protein was similar between the two ages (*Figure 2M*) despite higher *Rpl10a* RNA expression at E8.5 (FPKM: E8.5=1962.03; E10.5 = 1242.73), suggesting the involvement of post-transcriptional mechanisms. Transmission electron microscopic (TEM) analyses

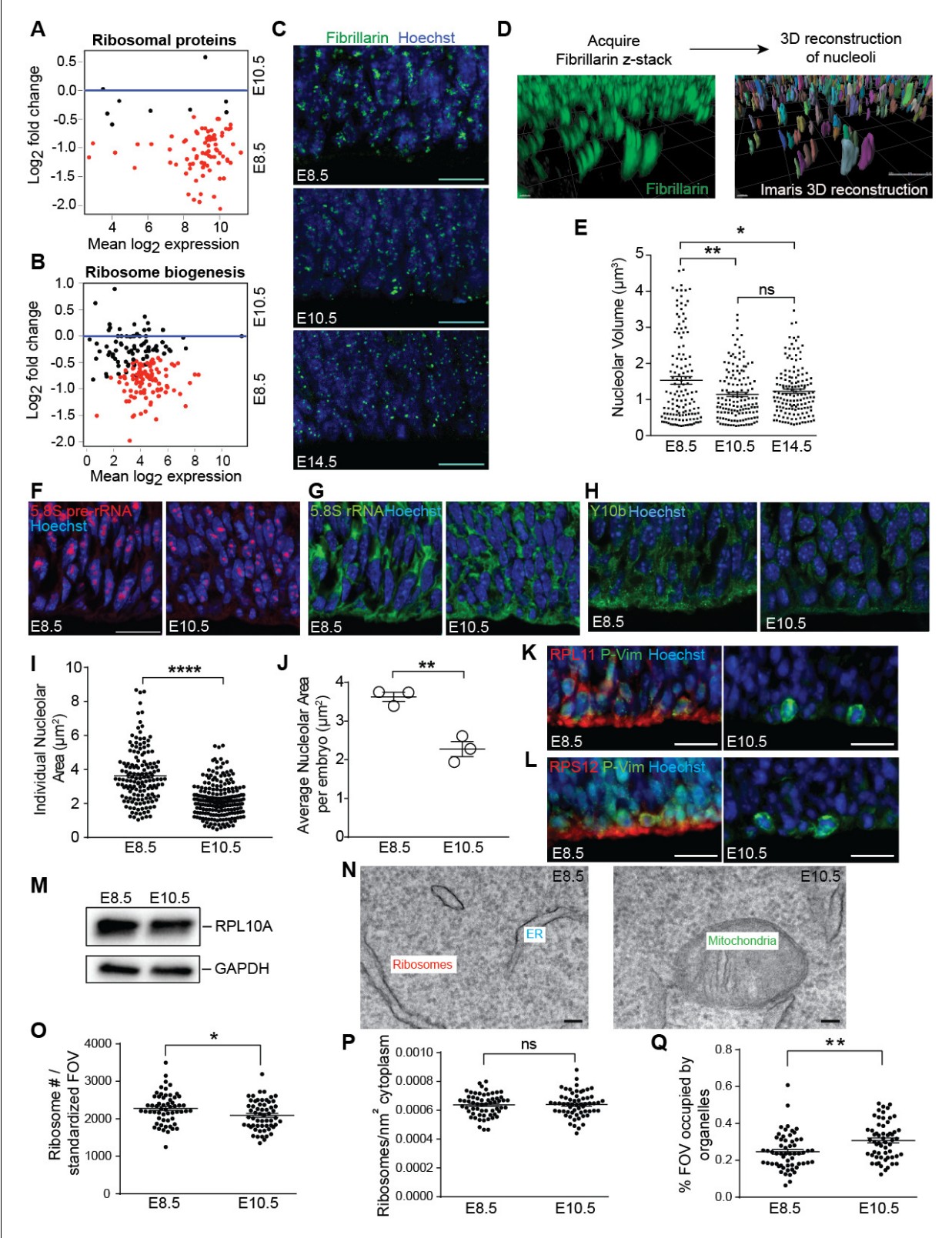

**Figure 2.** Ribosome biogenesis decreases from E8.5 to E10.5. (**A, B**) MA plot displaying genes encoding ribosomal proteins (**A**), ribosome biogenesis factors (**B**). Each dot represents a single gene. Red dots denote differentially expressed genes as identified by Cuffdiff (q < 0.05). Genes below blue line (y = 0) are enriched in E8.5. (**C**) Immunohistochemistry of the nucleolar protein Fibrillarin (green) in E8.5, E10.5 and E14.5 neuroepithelium. Scale bar = 20 µm. (**D**) Example of z-stack image of Fibrillarin staining (left) and 3D reconstruction of nucleoli using Imaris (right). (**E**) Quantification of

*Figure 2 continued on next page*

*Figure 2 continued*

nucleolar volume using Imaris. Each data point represents one nucleolus. *p≤0.05, **p≤0.01, Welch's ANOVA with Games-Howell post-hoc test. Sample size, E8.5: n = 135 from three embryos; E10.5: n = 139 from three embryos; E14.5: n = 146 from three embryos. (F, G) Representative images of fluorescent in situ hybridization of 5.8S pre-rRNA (F, red) and 5.8S total rRNA (G, green). (H) Y10b immunostaining shows higher levels of 5.8S rRNA in E8.5 than E10.5 neuroepithelium. Scale bar = 20 µm. (I) Quantification of 5.8S pre-rRNA signal shows larger nucleolar area in E8.5 compared to E10.5 neuroepithelium. Each data point represents one nucleolus. ****p≤0.0001, Welch's t-test. Sample size, E8.5: n = 150 from three embryos; E10.5: n = 202 from three embryos. (J) Average nucleolar area in E8.5 vs. E10.5 embryo. **p≤0.01, unpaired t-test, n = 3 embryos. (K, L) RPL11 (K, red) and RPS12 (L, red) were more highly expressed along the apical surface of E8.5 than E10.5 neuroepithelium. Phospho-Vimentin (P-Vim, green) labels dividing progenitors. Scale bar = 20 µm. (M) Immuoblotting shows similar expression of RPL10A between E8.5 and E10.5. (N) Representative images of TEM in neuroepithelial cells at E8.5 and R10.5. (O–Q) Quantification of TEM ribosomal number per standardized field of view (FOV), 78,736 nm$^2$, (O), ribosomal density in cytoplasm (P), and percent of the standard FOV occupied by membrane-bound organelles (Q). *p≤0.05, **p≤0.01, Unpaired t-test.

DOI: https://doi.org/10.7554/eLife.36998.004

revealed that E8.5 precursors had more ribosomes than E10.5 progenitors per field of view (*Figure 2N,O*). While ribosome density within the free cytoplasmic space was not different between these two ages (*Figure 2P*), more E10.5 cytoplasm than E8.5 cytoplasm was occupied by other organelles including endoplasmic reticulum, mitochondria, and Golgi (*Figure 2Q*), indicating an overall shift in organelle landscape at this age.

Gene expression analyses demonstrated the parallel downregulation of translational machinery from E8.5 to E10.5 progenitors (*Figure 3A*), including decreased expression of eukaryotic initiation factors (eIFs) such as EIF3η (*Figure 3B*). Activation of the growth promoting mTOR signaling pathway is linked to increased ribosome biogenesis and protein translation, a function mediated by the mTORC1 complex (*Laplante and Sabatini, 2012*). While *Mtor* expression itself was not changed from E8.5 to E10.5 (*Mtor* FPKM: E8.5=21.15; E10.5 = 25.91), components of the mTOR signaling pathway were differentially expressed and/or activated at these two ages (*Supplementary file 1*). For example, 4EBP1, a direct target of mTOR, showed increases in both expression and phosphorylation in the E8.5 neuroepithelium (*Figure 3C,D*). S6K1, a direct mTORC1 target that was similarly expressed at the two ages was also more highly phosphorylated at E8.5 compared to E10.5 (*Figure 3E*). Finally, S6 ribosomal protein, a substrate of S6K1, was more highly phosphorylated at E8.5 (*Figure 3F,G*). Taken together, these data demonstrate differential mTOR pathway activation in E8.5 compared to E10.5 neuroepithelium.

E8.5 neural progenitors showed higher $^{35}$S-methionine incorporation in vitro compared to E10.5 progenitors (counts per million cells, shown as E8.5 fold change normalized to E10.5 progenitors: *Expt. 1 = 2.0 fold; Expt. 2 = 1.4 fold; Expt. 3 = 1.1* fold), indicative of a higher protein synthesis rate in the younger forebrain progenitor cells. We next visualized actively elongating nascent polypeptides in vivo at the single-cell level using O-propargyl-puromycin (OPP; *Liu et al., 2012*) delivered maternally by intraperitoneal injection (*Figure 3H*). OPP incorporation was higher in E8.5 compared to E10.5 neuroepithelial cell bodies (*Figure 3I,J*), consistent with their larger nucleolar volumes and higher $^{35}$S-methionine incorporation. Collectively, these data demonstrate that presumptive forebrain progenitors have higher levels of ribosome biogenesis and protein synthesis compared to more mature progenitor cells of the developing forebrain.

## Downregulation of protein biosynthetic machinery matches AF and CSF proteomes

The early developing forebrain is bathed first by amniotic fluid (AF) and following neural tube closure, by cerebrospinal fluid (CSF). As neural progenitors can release signaling factors and membrane particles directly into the CSF (*Arbeille et al., 2015*; *Marzesco et al., 2005*), we tested the extent to which the changes observed in the forebrain transcriptome (*Figure 1*) reflected concurrent changes in the AF and CSF proteomes (*Chau et al., 2015*). We identified 691 proteins present in the AF and CSF that were also expressed by the developing forebrain neuroepithelium. Within this group of 691 proteins, the availability of 493 proteins matched gene expression patterns observed in the forebrain tissue: 395 proteins were enriched in E8.5 AF and more highly expressed by E8.5 neuroepithelium, *Figure 4A*, lower left quadrant; 98 proteins were enriched in E10.5 CSF and more highly expressed by E10.5 neuroepithelium, *Figure 4A*, upper right quadrant. Gene ontology analysis showed that, among proteins and genes enriched in E8.5 AF and neuroepithelium, the most highly represented

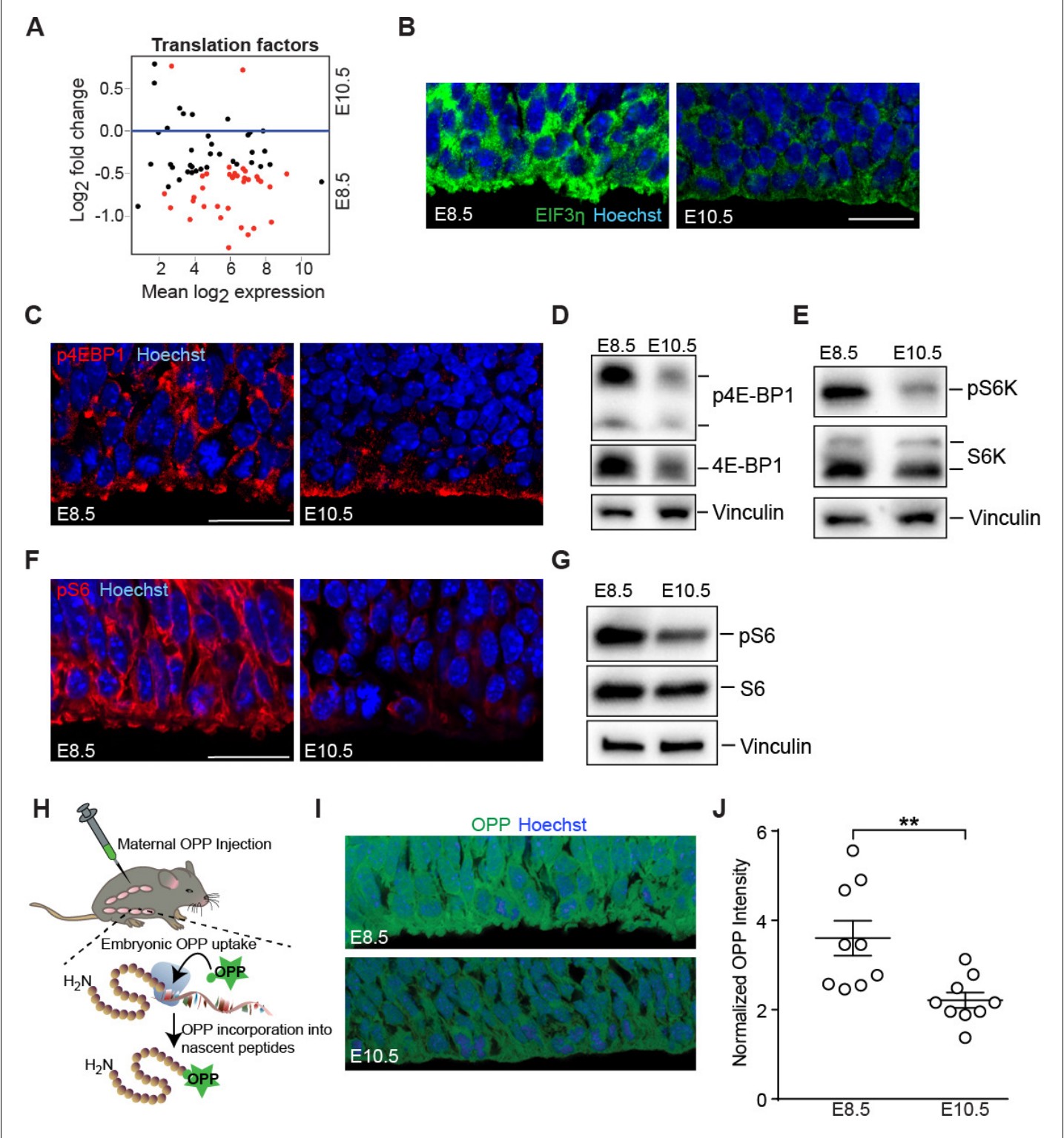

**Figure 3.** Downregulation of mTOR signaling pathway and decreased protein synthesis in E10.5 forebrain progenitors. (A) MA plot displaying genes encoding translation factors. Each dot represents a single gene. Red dots denote differentially expressed genes as identified by Cuffdiff (q<0.05). Genes below blue line (y = 0) are enriched in E8.5. (B) Immunostaining of developing forebrain progenitors shows higher expression of the translation initiation factor EIF3η (green) in E8.5 versus E10.5 neuroepithelium. Scale bar = 20 μm. (C) Immunostaining of developing forebrain neuroepithelium shows decreased phosphorylation of 4E-BP1 (red) in E10.5 neuroepithelium. Scale bar 20 μm. (D) Immunoblotting shows decreased expression and phosphorylation of 4E-BP1 at E10.5. (E) Immunoblotting shows decreased phosphorylation of S6K at E10.5. (F) Immunostaining of developing forebrain

*Figure 3 continued on next page*

Figure 3 continued

neuroepithelium shows decreased phosphorylation of ribosomal protein S6 (red) in E10.5 neuroepithelium. Scale bar 20 μm. (G) Immunoblotting shows decreased phosphorylation of ribosomal protein S6 at E10.5. (H) Schematic of OPP injection into pregnant dams and incorporation into translating polypeptides in the embryos. (I) OPP incorporation assay in E8.5 and E10.5 developing forebrain neuroepithelium. (J) Quantification of OPP fluorescence intensity using Image J shows decreased rate of protein synthesis at E10.5. **p≤0.01 Welch's t-test. For each age, n = 9 embryos from three litters.

DOI: https://doi.org/10.7554/eLife.36998.005

functional category was ribosomes/translation (*Figure 4B*). Further analysis revealed that nearly all ribosomal proteins and translation factors were enriched in both E8.5 AF and E8.5 neuroepithelium (*Figure 4C,D*), supporting the model that these fluid proteins can originate in the forebrain tissue. Not only were ribosomal proteins less abundant in E10.5 CSF, but many were no longer detected therein (*Figure 4E*; *Chau et al., 2015*). Together, these data demonstrate that the changes in the AF and CSF proteomes during early forebrain development match the downregulation of protein biosynthetic machinery in the adjacent neuroepithelium, thereby providing a developmental biomarker signature of concurrent cell biological changes in the developing forebrain.

## MYC modulates ribosome biogenesis in developing forebrain

In other cell types, the transcription factor MYC regulates genes encoding ribosomal proteins, proteins involved in ribosome biogenesis, and translation initiation and elongation factors (*van Riggelen et al., 2010*). Analyses of differentially expressed transcription factors between E8.5 and E10.5 neuroepithelium revealed that *Myc* expression was approximately ten-fold higher in E8.5 neuroepithelium (*Figure 5A*; *Myc* FPKM: E8.5=28.73, E10.5 = 2.76), suggesting MYC as a candidate regulator of ribosome biogenesis in the developing forebrain. There was no reciprocal compensatory suppression of *Mycn* or *Mycl* (*Mycn* FPKM: E8.5=28.03, E10.5 = 20.91; *Mycl* FPKM: E8.5=6.58, E10.5 = 10.34). We confirmed the high level of MYC expression in E8.5 neuroepithelium and its decreased expression in E10.5 neuroepithelium (*Figure 5B*, *Figure 5—figure supplement 1A–C*; see also *Shannon et al., 2018*). Once downregulated in E10.5 neural progenitors, MYC expression remained low throughout cerebral cortical development (*Figure 5B*).

GSEA demonstrated that many known MYC target genes were enriched in E8.5 compared to E10.5 neuroepithelium (*Figure 5C,D*; *Ben-Porath et al., 2008*; *Zeller et al., 2003*), and some of these target genes were associated with ribosome biogenesis and translation (e.g. *Ncl, Rps13*, and *Eif4e*). To test if interfering with MYC activity regulates ribosome biogenesis, we exposed wild type embryos to the MYC inhibitor, KJ-Pyr-9 (*Hart et al., 2014*) *in utero*, and observed smaller nucleoli compared to vehicle-injected controls (*Figure 5E*). In agreement with previous studies (*Davis et al., 1993*; *Zinin et al., 2014*), we confirmed that *Myc*-deficient embryos showed a triad of developmental defects including smaller size, neural tube closure defects, and developmental delay (*Figure 5—figure supplement 1D*). Nucleolar volume was also decreased in *Myc*-deficient embryos compared to developmentally stage-matched controls (*Figure 5F*).

MYC has important roles in cell cycle regulation (*Dang, 2013*). Therefore, its rapid downregulation by E10.5 was unexpected given that E10.5 represents a stage of continued progenitor proliferation and the start of forebrain neurogenesis. To determine the consequences of persistent MYC expression on cerebral cortical development, we genetically forced *MYC* expression by crossing StopFLMYC mice (*Calado et al., 2012*) with *Foxg1-Cre* (*Hébert and McConnell, 2000*) or *Nestin-Cre* (*Tronche et al., 1999*) mice (*Figure 5G*, *Figure 5—figure supplement 1E–G*). We purified Pax6-positive cortical progenitors at E13.5 (from *Nestin-Cre* cross, *Figure 5—figure supplement 1H*), and analyzed gene expression by RNA-seq. We identified 135 differentially expressed genes between WT and MYC-overexpressing (MYC-OE) embryos (q < 0.1), with 105 genes activated and 30 genes repressed in the MYC-OE progenitors (*Figure 5—figure supplement 1I*, *Supplementary file 2*). A cross-comparison between the 105 MYC activated genes with our early E8.5-E10.5 RNA-seq dataset (*Figure 1*) revealed 53 genes that were also enriched in E8.5 progenitors when MYC expression is naturally high (*Supplementary file 2*). Functional annotation clustering using DAVID revealed ribosomes as the most enriched gene category among the MYC-upregulated genes (*Figure 5H*, *Supplementary file 2*). GSEA further revealed that genes encoding ribosome components (*Figure 5I*), genes involved in ribosome biogenesis (*Figure 5J*), along with other known

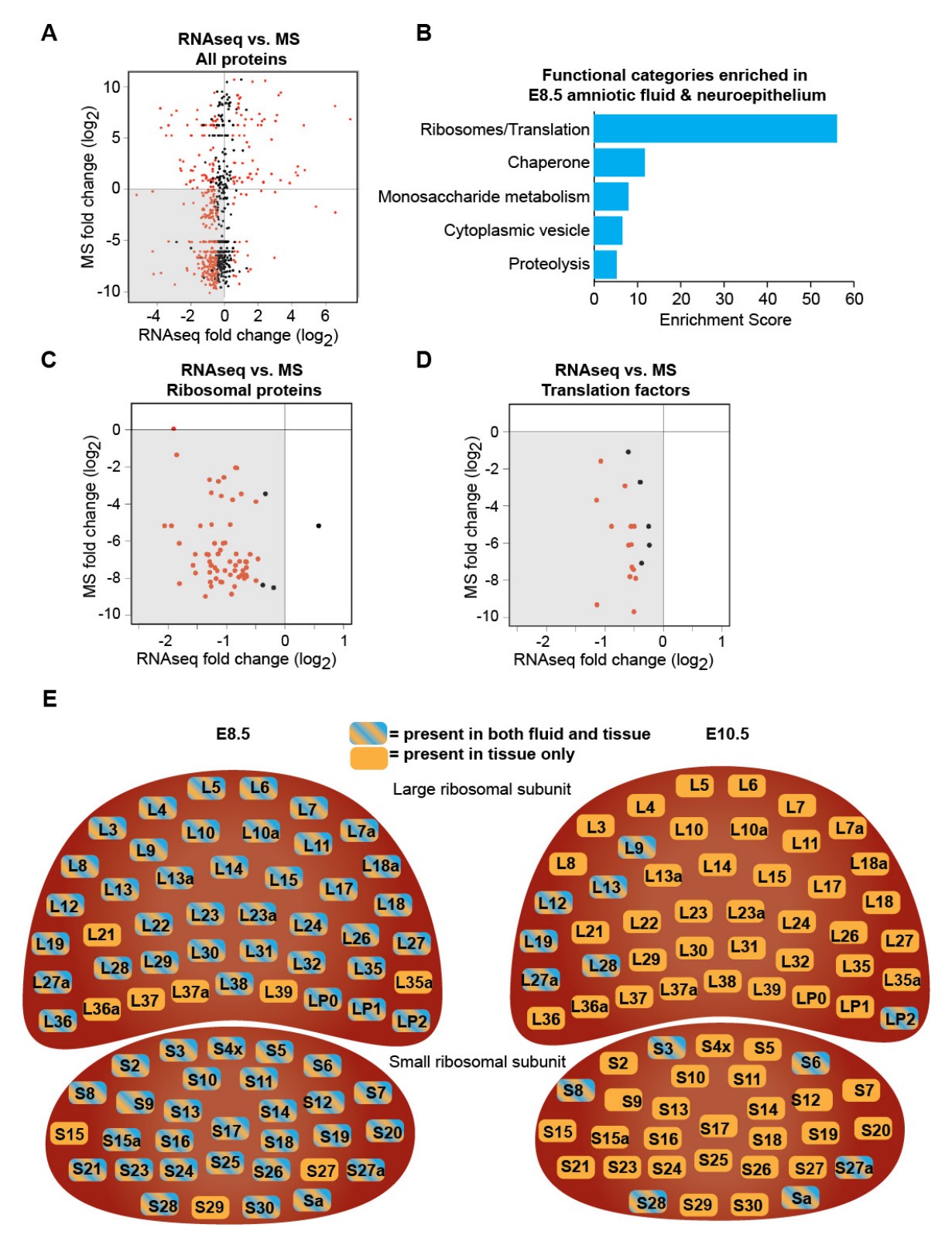

**Figure 4.** Downregulation of protein biosynthetic machinery during early forebrain development matches the AF and CSF proteomes. (**A**) Plot showing all proteins/genes that are detected in both AF/CSF and the neighboring neuroepithelium. Each dot represents a single protein/gene. Red dots denote differentially expressed genes between E8.5 and E10.5 epithelium (q<0.05). Genes left of x = 0 were enriched in E8.5 epithelium whereas proteins below y = 0 were enriched in E8.5 AF. Therefore, genes/proteins in lower left quadrant (shaded) were enriched in both E8.5 epithelium and
*Figure 4 continued on next page*

Figure 4 continued

AF. MS = mass spectrometry. (B) Functional annotation clustering of genes/proteins enriched in both E8.5 epithelium and AF (genes/proteins in shaded quadrant in (A)) shows that ribosomes/translation is the most overrepresented category. (C, D) Comparison of AF/CSF proteomes with neuroepithelium transcriptome showed that most ribosomal proteins and translation factors enriched in E8.5 AF were enriched in age-matched epithelium (shaded quadrants). (E) Schematics depicting the specific ribosomal protein subunits that were detected in E8.5 AF (left) and E10.5 CSF (right). Subunits with blue and orange were detected in both fluid and tissue, whereas those in orange were only detected in tissue.

DOI: https://doi.org/10.7554/eLife.36998.006

MYC target genes (*Figure 5—figure supplement 1J*; *Zeller et al., 2003*) were upregulated in the MYC-OE. Among the selective subset of ribosomal proteins that were differentially expressed in MYC-OE mice (q < 0.1), all were upregulated (*Figure 5K,L*, *Supplementary file 2*), even those subjected to less-stringent statistical parameters (q < 0.3, *Figure 5K*). These gene expression changes were accompanied by a modest increase in ribosome biogenesis in both *Foxg1-Cre* and *Nestin-Cre* MYC-OE mice (*Figure 5M,N*). Despite this upregulation of ribosome biogenesis and the expression of genes encoding translational machinery (*Figure 5—figure supplement 1K*), changes in protein synthesis at progenitor cell bodies were not consistently observed in either *Myc*-deficient or MYC-OE studies (data not shown). Taken together, these findings demonstrate that *Myc* modulates ribosome biogenesis in the developing forebrain, and that additional, as yet unidentified mechanisms participate in the regulation of protein biosynthesis at this developmental stage.

## Persistent MYC expression increases progenitor proliferation, leading to macrocephaly

Genes with known functions in regulating cerebral cortical neurogenesis were also upregulated in MYC-OE progenitors including *Insulin-like growth factor 2* (*Igf2*, *Figure 6—figure supplement 1A*, *Supplementary file 2*), which is typically not highly expressed by apical progenitors and is instead delivered by the CSF to regulate proliferation of progenitors (*Lehtinen et al., 2011*), and *Insulinoma-Associated 1* (*Insm1*, *Supplementary file 2*), which accelerates cortical development by promoting delamination of apical progenitor cells (*Farkas et al., 2008*; *Tavano et al., 2018*). The coordinated effects of MYC activation of several of these pathways resulted in a large brain phenotype that emerged by E14.5 in both *Foxg1-MYC* and *Nestin-MYC* mice (*Figure 6A–C*, *Figure 6—figure supplement 1B–D*), and was well defined by birth in *Nestin-MYC* mice (*Figure 6D–G*). No viable pups were recovered from the *Foxg1-Cre* cross (eight litters examined), indicating embryonic lethality between E14.5 and birth. This outcome may be due to the combinatorial effects of MYC overexpression and *Foxg1* heterozygosity (*Hébert and McConnell, 2000*), perhaps in tissues outside the brain. No differences in body weight were observed at P0 in *Nestin-MYC* mice (body weight [g]± SEM: WT = 1.35 ± 0.03, n = 16; MYC-OE = 1.33 ± 0.02, n = 15; unpaired t-test, p=0.57). By P8, MYC-OE and control brains were similar in size (brain weight [g]±SEM: WT = 0.38 ± 0.01, n = 9; MYC-OE = 0.39 ± 0.02, n = 6; p=0.72, unpaired t-test). However the MYC-OE mice had much smaller body size (body weight [g]±SEM: WT = 4.86 ± 0.09, n = 13; MYC-OE = 3.20 ± 0.22, n = 9; p<0.0001, Welch's t-test), leading to a sustained difference in their brain-body ratio (brain weight/ body weight: WT: 0.081 ± 0.001, n = 9; MYC-OE: 0.127 ± 0.006, n = 6; p=0.0003, Welch's t-test).

While no tumors were observed at the ages examined in this study, histological analyses suggested that MYC-OE by the *Nestin* promoter increased the size of the entire brain. A two-hour BrdU pulse delivered at E15.5 showed a larger proportion of Pax6-positive apical progenitors in S-phase in MYC-OE mice (*Figure 6H*), contributing to increased cortical thickness in MYC-OE mice by birth (*Figure 6I,J*). MYC-OE cortices had increased Cux1-positive staining cells destined for the upper layers of the cerebral cortex (*Figure 6K,L*), which contributed to the increased overall number of cells in the cerebral cortex (cell number±SEM: WT: 2,348 ± 199.3; MYC-OE: 2,518 ± 174.3, n = 4, p=0.07, paired t-test). On the other hand, no difference was observed in the number of Ctip2-positive lower layer neurons (cell number±SEM: WT: 651.5 ± 35.4; MYC-OE: 639 ± 34.9, n = 4, p=0.82, paired t-test). Together, our findings support the model that MYC overexpression in the *Nestin* lineage affects multiple pathways and that their convergence influences the development of the brain and the entire organism.

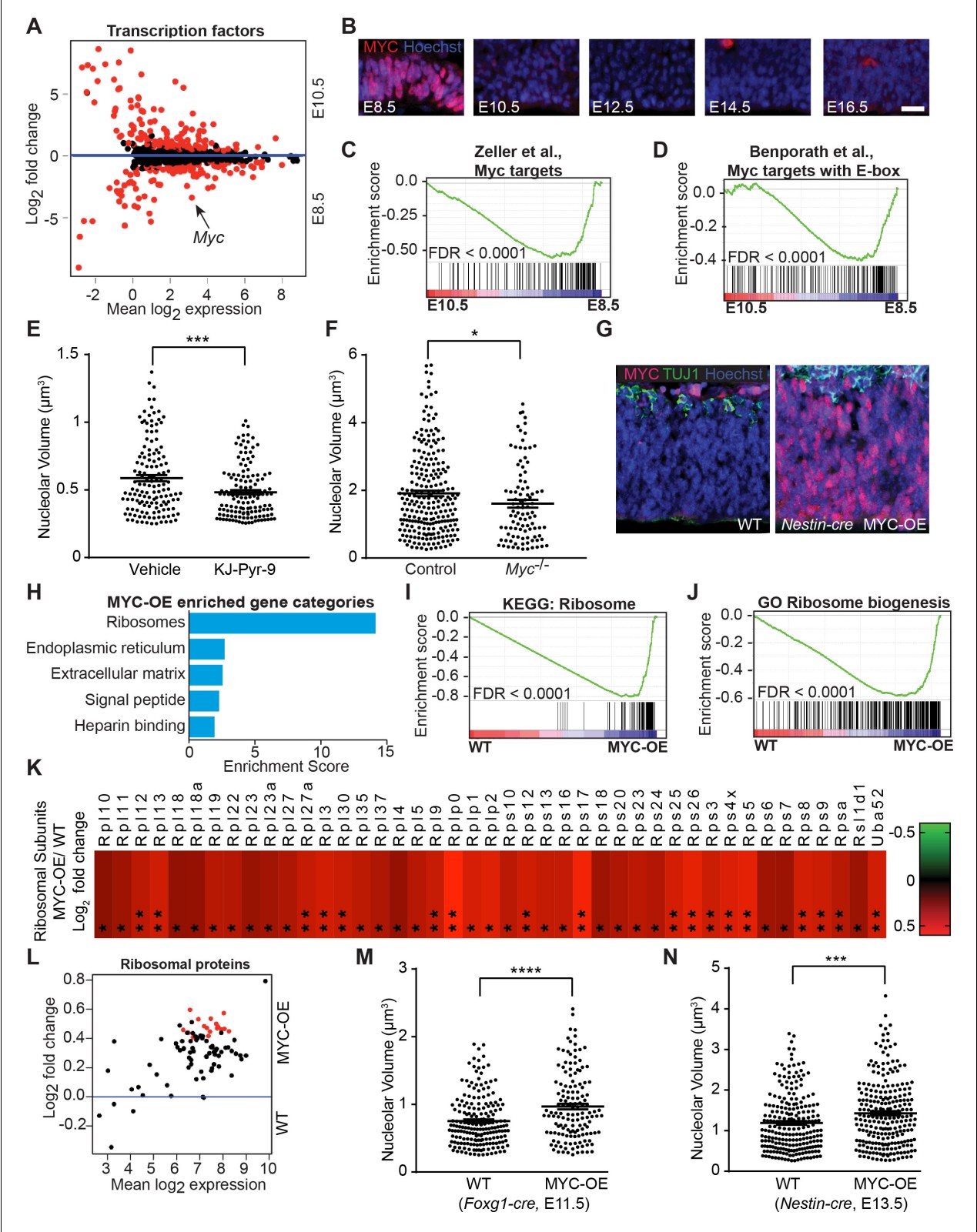

**Figure 5.** MYC modulates ribosome biogenesis in the developing forebrain. (**A**) MA plot displaying genes encoding transcription factors in E8.5 and E10.5 neuroepithelium. Each dot represents a single gene. Red dots denote differentially expressed genes identified by Cuffdiff (q<0.05). Genes below blue line (y = 0) enriched in E8.5. *Myc* (arrow) expression is ~10 fold higher in E8.5 epithelium (FPKM: E8.5=28.73, E10.5 = 2.76). (**B**) MYC expression was enriched in E8.5 neuroepithelium. Once downregulated at E10.5, MYC expression remained low throughout cortical development. Scale bar = 20 μm. *Figure 5 continued on next page*

*Figure 5 continued*

(C, D) GSEA of E8.5 versus E10.5 neuroepithelium for gene sets containing genes up-regulated by MYC and whose promoters are bound by MYC (C), and E-box containing MYC target genes (D). Broad Institute Molecular Signatures Database Identifiers: DANG_MYC_TARGETS_UP (C), BENPORATH_MYC_TARGETS_WITH _EBOX (D). Each line represents a single gene in the gene set; genes on the right side enriched in E8.5. (E) Quantification of nucleolar volume of E8.5 embryos treated with vehicle control or KJ-Pyr-9 for 24 hr. Each data point represents one nucleolus. ***$p \leq 0.001$, Welch's t-test. Sample size, vehicle: n = 140 from three embryos; KJ-Pyr-9: n = 140 from three embryos. (F) Quantification of nucleolar volume of $Myc^{-/-}$ compared to controls (wild type and heterozygous littermates) in E8.5 neuroepithelium. *$p \leq 0.05$ Unpaired t-test. Sample size, controls: n = 238 from five embryos; $Myc^{-/-}$: n = 97 from two embryos. (G) Immunostaining shows overexpression of MYC (red) in the developing cortex of E12.5 MYC-OE (right) embryos from the *Nestin-cre* x StopFLMYC cross. TUJ1 (green) staining labels neurons. (H) Functional annotation clustering of the 105 MYC-OE enriched genes shows overrepresentation of genes encoding ribosome constituents. The top five enriched functional clusters are shown. (I, J) GSEA of WT versus MYC-OE apical progenitors for gene sets involved in ribosome biogenesis. Broad Institute Molecular Signatures Database Identifiers: KEGG_RIBOSOME (I), and GO_RIBOSOME_ BIOGENESIS (J). Each line represents a single gene in the gene set, genes on the right side are enriched in MYC-OE. (K) Heatmap of the 43 ribosomal protein genes that are differentially expressed between MYC-OE and WT apical progenitors (* q < 0.3, **q < 0.1). All ribosomal proteins are more highly expressed in MYC-OE. Red and green indicate relatively higher and lower expression, with gene FPKM values log2 transformed. (L) MA plot displaying genes encoding ribosomal proteins in E13.5 apical progenitors. Each dot represents a single gene. Red dots denote differentially expressed genes as identified by Cuffdiff (q<0.1). Genes above blue line (y = 0) are enriched in MYC-OE. (M) Quantification of nucleolar volume of WT and MYC-OE (*Foxg1-cre* driven) forebrain progenitors at E11.5. Each data point represents one nucleolus. ****$p \leq 0.0001$, Welch's t-test. Sample size, WT: n = 194 from four embryos; MYC-OE: n = 144 from three embryos. (N) Quantification of nucleolar volume of WT and MYC-OE (*Nestin-cre* driven) apical progenitors at E13.5. Each data point represents one nucleolus. ***$p \leq 0.001$, Welch's t-test. Sample size, WT: n = 234 from five embryos; MYC-OE: n = 248 from five embryos.

DOI: https://doi.org/10.7554/eLife.36998.007

The following figure supplement is available for figure 5:

**Figure supplement 1.** MYC expression and mouse models.

DOI: https://doi.org/10.7554/eLife.36998.008

## Discussion

Our study reveals major changes in expression of the protein biosynthetic pathway during early specification of the mammalian forebrain. This work (1) demonstrates that enhanced biogenesis of ribosomes and protein synthetic machinery serve as transcriptional and cell biological signatures defining early forebrain precursor cells; (2) reveals that the changing proteomes of AF and CSF provide a biomarker signature that matches the concurrent, normal development of the adjacent forebrain; (3) identifies MYC as a contributor to the regulation of ribosome biogenesis in the developing forebrain; and (4) shows that persistent MYC expression leads to increased ribosome biogenesis, enhanced cortical progenitor proliferation, and macrocephaly. We conclude that, as in other stem cells, neural progenitor cells dynamically regulate protein biosynthetic machinery to meet their changing needs, and that this process is regulated in part by MYC.

The DNA transcriptome is an essential starting point for our understanding of tissue regionalization, patterning, and individual cell identities in the mammalian central nervous system. Nevertheless, not all mRNAs are selected for protein translation, and our discovery of temporal regulation of the protein biosynthetic machinery during early specification of the forebrain uncovers a new layer of regulation fundamental to the early construction of the brain. Regulation of the protein biosynthetic machinery provides a tunable molecular program harnessed by cells to guide transitions between stem cell states (*DeBoer et al., 2013*; *Fujii et al., 2017*; *Khajuria et al., 2018*; *Kraushar et al., 2016*; *Sanchez et al., 2016*; *Scognamiglio et al., 2016*). Cell cycle in the forebrain lengthens over the course of development (*Caviness and Takahashi, 1995*). As such, the higher rates of ribosome biogenesis and protein synthesis observed in neurectodermal precursors relative to post-neurulation progenitors are consistent with a model in which rapidly dividing cells synthesize more proteins to support their proliferation (*Buszczak et al., 2014*). Genes such as *Pelo* and *Abce1* are downregulated in E10.5 progenitors (*Supplementary file 1*), suggesting that additional levels of translational control including ribosome recycling may be engaged during this developmental time window (*Dever and Green, 2012*).

Disruptions in ribosome structure and function are linked to a number of genetically inherited ribosomopathies such as Diamond-Blackfan anemia (*Boria et al., 2010*; *Choesmel et al., 2007*; *Ebert and Lipton, 2011*). Nucleolar size, ribosome biogenesis, and protein translation have been implicated in aging and longevity (*Buchwalter and Hetzer, 2017*; *Tiku et al., 2016*). In the central

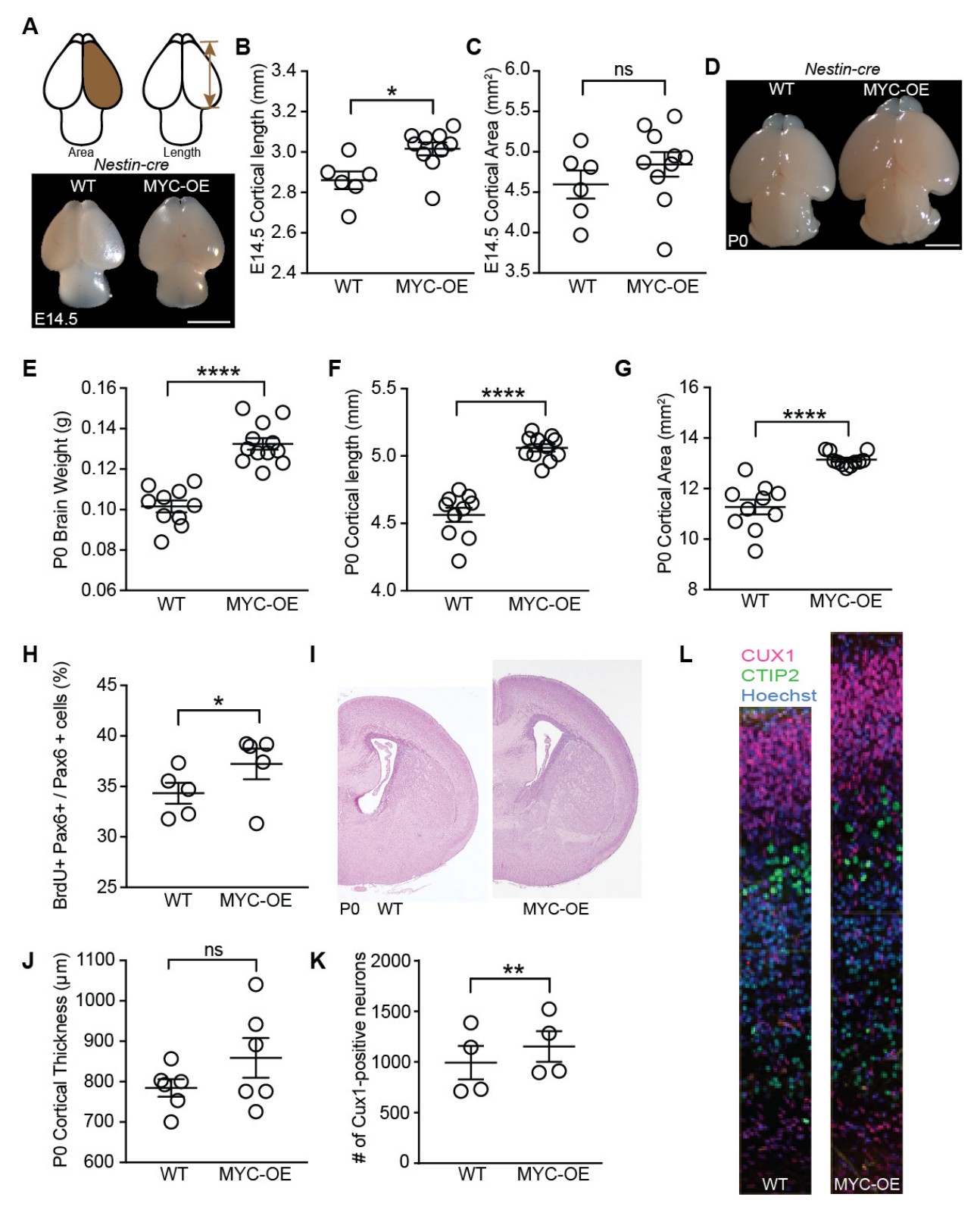

**Figure 6.** Persistent MYC expression in cortical progenitors leads to macrocephaly. (A) Representative images of E14.5 brains from WT and MYC-OE from the *Nestin-cre* x StopFLMYC cross. Scale bar = 2 mm. (B) Quantification of E14.5 cortical length (olfactory bulb excluded). *p≤0.05, unpaired t-test, WT: n = 6 from two litters, MYC-OE: n = 10 embryos from two litters. (C) Quantification of E14.5 cortical area. Cortical area of one hemisphere was measured (olfactory bulb excluded). p>0.05, unpaired t-test, WT: n = 6 from two litters, MYC-OE: n = 10 embryos from two litters. (D)
*Figure 6 continued on next page*

*Figure 6 continued*

Representative images of P0 brains from WT and MYC-OE from the *Nestin-cre* x StopFLMYC cross. Scale bar = 2 mm. (**E**) Quantification of P0 brain weight. Olfactory bulb, medulla and pons were excluded from measurements. ****p≤0.0001, unpaired t-test, No outliers, WT: n = 10 pups from three litters, MYC-OE: n = 12 pups from three litters. (**F**) Quantification of P0 cortical length as in (**B**). ****p≤0.0001, unpaired t-test, outlier excluded by ROUT method, WT: n = 10 pups from three litters, MYC-OE: n = 11 pups from three litters. (**G**) Quantification of P0 cortical area as in (**C**). ****p≤0.0001, Welch's t-test, outlier excluded by ROUT method, WT: n = 10 pups from three litters, MYC-OE: n = 11 pups from three litters. (**H**) Percent PAX6-positive progenitors that were also BrdU-positive after a 2 hr BrdU pulse at E15.5. *p≤0.05, Welch's t-test, n = 5 embryos from three litters. (**I**) Representative H and E staining of WT and MYC-OE forebrain at P0. (**J**) Quantification of cortical thickness of P0 cortex. Thickness is measured from the ventricular surface to the pial surface in the dorsal-lateral cortex. p>0.05, unpaired t-test, n = 6 pups from five litters. (**K**) MYC-OE had increased number of CUX1-positive upper layer neurons at P0. **p≤0.01; paired t-test, n = 4 litters, 1–2 pairs of embryos per litter were quantified. (**L**) Examples of 100 μm wide cortical columns at P0 used for cell counting. CUX1: upper layer neurons (red), CTIP2: lower layer neurons (green).
DOI: https://doi.org/10.7554/eLife.36998.009

The following figure supplement is available for figure 6:

**Figure supplement 1.** MYC overexpression in neural progenitors driven by *Foxg1-cre* leads to slightly longer cortex at E14.5.
DOI: https://doi.org/10.7554/eLife.36998.010

nervous system, disruptions in mRNA processing and translation can eventually impair the form and function of delicate neural circuitry. Changes in mRNA binding proteins are linked to neurodevelopmental disorders including autism spectrum disorder (*Kraushar et al., 2014*; *Popovitchenko et al., 2016*), and stem cell-derived neural progenitors from schizophrenia patients have altered levels of protein synthesis (*Topol et al., 2015*). During later stages of neurogenesis in the cerebral cortex, subcellular transport of mRNA by binding proteins including FMRP (Fragile-X mental retardation protein) ferry mRNA to sites of local translation in more polarized cells (*Kwan et al., 2012*; *Pilaz et al., 2016*; *Pilaz and Silver, 2017*). It is tempting to speculate that in the developing forebrain, the assembly of specialized ribosomes could enable unique or localized translation in developing precursor cells, fine-tuning cellular identities and tailoring individualized developmental programs as in other tissues (*Bortoluzzi et al., 2001*; *Fujii et al., 2017*; *Shi et al., 2017*; *Simsek et al., 2017*).

While ribosomal protein expression and ribosome biogenesis decrease as the embryo develops, we did not observed any difference in cytoplasmic ribosome density from EM analysis (*Figure 2*). This might be due to the long half-life of ribosomes (*Hirsch and Hiatt, 1966*; *Nikolov et al., 1983*), and thus ribosomes generated earlier at E8.5 would likely still be present at E10.5. Using OPP and methionine incorporation, we provided evidence that protein synthesis is also downregulated as the embryo develops. However, we do not know whether differential protein synthesis is driven by changes in ribosome biogenesis or by overall changes in transcription dynamics. Furthermore, OPP was administered intraperitoneally into the pregnant dams, and it is possible that availability of OPP to the embryonic progenitors might be different between E8.5 and E10.5. Indeed, increased ribosomal protein expression does not always result in increased translation because not all ribosomal proteins are associated with polysomes (*Kraushar et al., 2015*). It is possible that some ribosomal proteins perform extraribosomal functions independent of translation (*Warner and McIntosh, 2009*; *Zhou et al., 2015*). For instance, RPL11 is recruited to the promoter regions of p53 target genes during nucleolar stress to promote p53 transcriptional activity (*Mahata et al., 2012*). Therefore, additional evidence is needed to confirm that changes in ribosome biogenesis directly cause differential protein synthesis in the early neural progenitors.

Neural progenitors depend on their adjacent fluid environment for appropriate fluid pressure and instructive signals (*Lun et al., 2015*). Developing neural tissue also releases membrane bound vesicles into the adjacent fluid environment (*Cossetti et al., 2014*; *Marzesco et al., 2005*). We found that the protein biosynthetic changes occurring in the forebrain neuroepithelium were reflected in the proteomic content of the adjacent AF and CSF (*Figure 4*). The CSF is commonly sampled for biomarkers of neurologic diseases. Our data demonstrate that during early forebrain development, the proteomic signature of the early brain fluids provides a biomarker signature of the normal, healthy forebrain, opening a new 'window' into this stage of early brain development. Whether the ribosomal and translational machinery found in the AF and CSF are equipped to actively synthesize proteins within the fluid environment remains to be elucidated. Alternatively, the fluids might serve as a channel for intercellular transfer of ribosomes and other proteins

(*Cossetti et al., 2014*; *Court et al., 2008*). Future studies will also reveal whether maturation-associated release of protein biosynthetic machinery into the developing brain fluids is an active or passive process, and whether this process shares features with membrane shedding that occurs in other cell types, such as at the maturing red blood-cell surface (*Gautier et al., 2016*).

The swift downregulation of MYC following neurulation could be due to chromatin modifications, epigenetic mechanisms, and/or inhibition of RNA polymerase II elongation. *Myc*-deficiency (*Kerosuo and Bronner, 2016*) as well as ground-level changes in DNA methylation, histone modifications, and nucleosome positioning are associated with NTD (reviewed in *Greene and Copp, 2014*; *Wilde et al., 2014*). Cross-referencing our data (*Figure 1*) with NTD Wiki, a repository of genes required for neurulation (www.ntdwiki.wikispaces.com), revealed that a number of MYC targets are associated with NTD (data not shown). Complex gene-environment interactions have long been appreciated to underlie NTD. Despite modern successes in reducing the incidence of NTD by dietary fortification (e.g. folate) and increased awareness of adverse consequences of maternal exposures (e.g. alcohol and drug use) on the developing fetus, NTD continue to represent one of the most common birth defects worldwide (*Wallingford et al., 2013*). Neurulation varies along the anterior-posterior axis, and specific cell types (e.g. hinge points, neural fold cells) have distinct roles in this process (*Massarwa et al., 2014*). Thus, while our study investigated anterior forebrain development, variation in expression of the protein biosynthetic machinery along the anterior-posterior and dorsal-ventral/medial-lateral axes could differentially affect neurulation along the entire body axis. Overall, the identification of molecular pathways regulating protein biosynthetic machinery during neurulation may provide new opportunities to seek answers to these complex conditions.

Aberrant regulation of the signaling pathways examined in this study in cortical progenitors are associated with cortical overgrowth syndromes such as hemimegalencephaly, a brain malformation characterized by unilateral enlargement of one hemisphere (*D'Gama et al., 2017*; *Poduri et al., 2012*). Increased *MYC* expression has been reported in hemimegalencephaly (*Yu et al., 2005*), though to our knowledge, mutations in *MYC* itself have not been shown to drive the pathogenesis of this malformation.

While perhaps best known for its role as an oncogene, we did not observe any cortical tumors in *Nestin:MYC* brains. Context-dependent effects of MYC have been reported, with age- and tissue-dependent effects on cellular phenotypes including proliferation and cell growth (*Gabay et al., 2014*; *Zinin et al., 2014*). The tumorigenic consequences of persistent *MYC* expression of this model emerged later in adult mice as choroid plexus carcinoma and ciliary body medulloepithelioma (*Shannon et al., 2018*), exposing the select vulnerability of certain subtypes of epithelial cells in the *Nestin* lineage to tumorigenesis. Such selectivity of MYC-associated pathologies may be determined by the epigenetic landscape of differentiated cells in adult tissues. MYC may act as a universal amplifier of expressed genes, promoting proliferation in already dividing cells (*Lin et al., 2012*; *Nie et al., 2012*). However, in more differentiated cells, target genes may be confined to heterochromatin and inaccessible to MYC (*Kress et al., 2015*). Certain cell types may also require a genetic double-hit such as concomitant *p53*-deficiency in the cortex (*Momota et al., 2008*), or particular gene-environment triggers, for transformation.

Overall, cellular identity and health reflect the net equation between a cell's transcriptional and translational output (*Buszczak et al., 2014*; *Fujii et al., 2017*; *Holmberg and Perlmann, 2012*; *Khajuria et al., 2018*; *Sanchez et al., 2016*). These processes require multiple regulatory steps that are vulnerable to disruptions accumulating from cell-intrinsic genetic programs, and/or cell-extrinsic environmental cues. In the developing brain, environmental signals can entail disturbances of local gradients diffusing through tissues (e.g. *Toyoda et al., 2010*) or altered delivery of growth-promoting factors by the adjacent AF or CSF (*Chau et al., 2015*; *Lehtinen et al., 2011*). All of these signaling activities are susceptible to exogenous maternal exposures including illness, substance abuse, and environmental toxins. Thus, our findings provide a new paradigm for understanding brain development through investigation of molecular pathways regulating the biosynthetic machinery in forebrain progenitors.

# Materials and methods

## Key resources table

| Reagent type (species) or resource | Designation | Source or reference | Identifiers | Additional information |
|---|---|---|---|---|
| Strain, strain background (*Mus Musculus*) | Gt(ROSA)26Sor[tm13(CAG-MYC,-CD2*)Rsky] (referred as StopFLMYC) | The Jackson Laboratory (Bar Harbor, ME) | MGI:5444670 | Maintained on a C57Bl/6 background |
| Strain, strain background (*Mus Musculus*) | Tg(Nes-cre)1Kln (referred as *Nestin-cre*) | The Jackson Laboratory | MGI:2176173 | Maintained on a C57Bl/6 background |
| Strain, strain background (*Mus Musculus*) | Foxg1[tm1(cre)Skm] (referred as *Foxg1-cre*) | The Jackson Laboratory | MGI:1932522 | Maintained on a C57Bl/6 background |
| Strain, strain background (*Mus Musculus*) | Myc-deficient mice (*c-myc$^{-/-}$*) | Provided by Troy Baudino | *Baudino et al. (2002)* | Maintained on a C57Bl/6 background |
| Strain, strain background (*Mus Musculus*) | CD-1 IGS Mouse (referred as CD-1) | Charles River (Wilmington, MA) | Strain code: 022 | Wildtype timed pregnant mice |
| Antibody | Rabbit anti-4E-BP1 | Cell Signaling (Danvers, MA) | 9644 | 1:1000 |
| Antibody | Mouse anti-5.8S ribosomal RNA [Y10B] | Abcam (United Kingdom) | ab171119 | 1:50; antigen retrieval with steaming in citric acid |
| Antibody | Mouse anti-ACTB | Cell Signaling | 12262 | 1:2000 |
| Antibody | Rat anti-BrdU | Biorad (Hercules, CA) | MCA2060 | 1:200; antigen retrieval with steaming in citric acid |
| Antibody | Rabbit anti-MYC | Abcam | ab32072 | 1:100 for IHC, antigen retrieval with steaming in citric acid; 1:2000 for WB |
| Antibody | Rat anti-CTIP2 | Abcam | ab18465 | 1:200 |
| Antibody | Rabbit anti-CUX1 | Santa Cruz Biotechnology (Dallas, TX) | sc13024 | 1:200 |
| Antibody | Mouse anti-EIF3η | Santa Cruz Biotechnology | sc137214 | 1:100; antigen retrieval with steaming in citric acid |
| Antibody | Mouse anti-Fibrillarin | Abcam | ab4566 | 1:250; antigen retrieval with steaming in citric acid |
| Antibody | Mouse anti-GAPDH | Cell Signaling | 97166 | 1:1000 |
| Antibody | Rabbit anti-p4E-BP1 | Cell Signaling | 2855 | 1:200 for IHC; 1:1000 for WB |
| Antibody | Rabbit anti-PAX6 | Biolegend (San Diego, CA) | 901301 | 1:100; antigen retrieval with steaming in citric acid; 1:1000 for FACS |
| Antibody | Rabbit anti-pS6 | Cell Signaling | 5364 | 1:200 for IHC; 1:1000 for WB |
| Antibody | Rabbit anti-pS6K | Cell Signaling | 9234 | 1:1000 |
| Antibody | Mouse anti-pVimentin | Enzo Bioscience (Farmingdale, NY) | ADI-KAM-CC249-E | 1:400 |
| Antibody | Mouse anti-RPL10A | Novusbio (Littleton, CO) | H00004736-M01 | 1:500 |
| Antibody | Rabbit anti-RPL11 | Santa Cruz Biotechnology | sc50363 | 1:50 |
| Antibody | Rabbit anti-RPS12 | Proteintech (Chicago, IL) | 16490–1-AP | 1:50 |
| Antibody | Rabbit anti-S6 | Cell Signaling | 2217 | 1:1000 |
| Antibody | Rabbit anti-S6K | Cell Signaling | 9202 | 1:1000 |
| Antibody | Mouse anti-TUJ1 | Biolegend | 801202 | 1:100 for IHC; 1:1000 for FACS |
| Antibody | Rabbit anti-Vinculin | Cell Signaling | 13901 | 1:1000 |

*Continued on next page*

*Continued*

| Reagent type (species) or resource | Designation | Source or reference | Identifiers | Additional information |
|---|---|---|---|---|
| Recombinant DNA reagent | Quaser 570 coupled 5.8S pre-rRNA FISH probe | Provided by Debra Silver | | 1:200 |
| Recombinant DNA reagent | Quaser 670 coupled 5.8S total rRNA FISH probe | Provided by Debra Silver | | 1:200 |
| Commercial assay or kit | RecoverAll Total Nucleic Acid Isolation Kit for FFPE | Ambion (Foster City, CA) | AM1975 | Manufacturer's protocol |
| Commercial assay or kit | Ovation RNA-Seq System V2 | Nugen (San Carlos, CA) | 7102 | Manufacturer's protocol |
| Commercial assay or kit | Ovation Ultralow System V2 1–16 | Nugen | 0344 | Manufacturer's protocol |
| Commercial assay or kit | TruSeq RNA Library Prep Kit v2 | Illumina (San Diego, CA) | RS-122 | Manufacturer's protocol |
| Commercial assay or kit | Rneasy Micro Kit | Qiagen (Germany) | 74004 | Manufacturer's protocol |
| Commercial assay or kit | Pierce BCA Protein Assay Kit | Thermo Fisher Scientific (Waltham, MA) | 23227 | |
| Commercial assay or kit | Click-iT plus OPP protein synthesis assay kit | Thermo Fisher Scientific | C10456 | |
| Chemical compound, drug | O-propargyl-puromycin (OPP) | Life Technologies (Carlsbad, CA) | C10459 | IP injection, dosage: 50 mg/kg |
| Chemical compound, drug | KJ-Pyr-9 | Tocris (United Kingdom) | 5306 | IP injection, dosage: 10 mg/kg |
| Chemical compound, drug | $^{35}$S-Methionine | Perkin Elmer (Waltham, MA) | NEG709A | 51 μCi |
| Chemical compound, drug | 5-Bromo-2'-deoxyuridine (BrdU) | Sigma (St. Louis, MO) | B5002 | IP injection, 50 mg/kg |
| Software, algorithm | TopHat | https://ccb.jhu.edu/software/tophat/index.shtml | v2 | RNAseq analysis |
| Software, algorithm | Cufflinks | http://cole-trapnell-lab.github.io/cufflinks/ | v2 | RNAseq analysis |
| Software, algorithm | DAVID | https://david.ncifcrf.gov/ | v6.7, 6.8 | RNAseq analysis |
| Software, algorithm | GSEA | http://software.broadinstitute.org/gsea/index.jsp | v2 | RNAseq analysis |
| Software, algorithm | R Studio | Rstudio, Inc. | v0.99 | RNAseq analysis |
| Software, algorithm | Prism | GraphPad | v7 | Statistical analysis |
| Software, algorithm | FIJI (Image J) | https://fiji.sc/# | v1 | Image analysis |
| Software, algorithm | Imaris | Bitplane | | Image analysis |

## Mice

Timed pregnant CD1 dams were obtained from Charles River Laboratories. *Myc*-deficient mice (*Baudino et al., 2002*) were maintained in a C57BL/6J genetic background. StopFLMYC mice (JAX: 020458) were maintained in a C57BL/6J genetic background and crossed with *Nestin-cre* line (JAX: 003771) or *Foxg1-cre* line (JAX: 004337) to generate MYC-OE mice, in which human *MYC* transgene is selectively expressed in neural progenitor cells. All analyses were carried out using male and female mice. All animal experimentation was carried out under protocols approved by the IACUC of Boston Children's Hospital.

## E8.5 and E10.5 forebrain epithelium RNAseq

Forebrain epithelium at E8.5 and E10.5 was dissected as described (*Chau et al., 2015*). Each sequenced sample comprised forebrain epithelial tissues pooled across one litter. Total RNA was isolated using the RNeasy Micro Kit (Qiagen), converted to cDNA, and preamplified using the Ovation RNA-seq System V2 (NuGEN) following the manufacturer's instructions. cDNA was converted to

Illumina paired-end sequencing libraries following the standard protocol (TruSeq v2) and sequenced on a Illumina HiSeq 2000 instrument to a depth of ~20–60 million pass-filter reads per library, after standard quality control filters. The 50 base pair paired-end reads were mapped to the UCSC mm9 mouse reference genome using TopHat v2, and fragments per kilobase per million reads (FPKM) values were estimated using cufflinks v2, and differentially expressed genes (DEG) were identified using cuffdiff v2 with q value < 0.05 (*Trapnell et al., 2012*).

## FACS of neural progenitors

E13.5 dorsal telencephalon was microdissected, avoiding the lateral ganglionic eminence and structures ventral to it. The cortex was separated from the meninges, and cortices from samples of the same genotype were pooled and sliced into small, uniformly sized pieces. Tissues were digested with 2.5% Trypsin (Invitrogen, Carlsbad, CA), then dissociated into single cells by repeated pipetting. Cells were fixed in 4% paraformaldehyde (PFA), incubated with primary antibodies, and then secondary antibodies. Each step was carried out in 4℃ for 30 mins, with rotation. RNAsin (NEB, Ipswich, MA) was added to buffers to prevent RNA degradation (*Hrvatin et al., 2014*). Cells were sorted using FACS Aria IIU (BD).

Antibodies: Rabbit anti-PAX6 (Biolegend 901301, 1:1000), Mouse anti-TUJ1 (Biolegend 801202, 1:1000)

## E13.5 neural progenitor RNAseq

RNA was extracted from sorted neural progenitors using RecoverAll Total Nucleic Acid Isolation Kit (Ambion), then reverse transcribed into cDNA and pre-amplified using Ovation RNA-Seq System V2 (Nugen 7102). Libraries were prepared using Ovation Ultralow System V2 1–16 (Nugen 0344), and sequenced (Illuminia HiSeq 2500) to a depth of ~25–40 million reads per library. The 50 base pair single-end reads were mapped to the UCSC mm10 mouse reference genome using TopHat v2, FPKM values were estimated using cufflinks v2, and DEG were identified using cuffdiff v2 with q value < 0.1 (*Trapnell et al., 2012*).

## RNAseq data analysis

All analyses were performed using genes with FPKM >1, which we considered as the threshold of expression. Hierarchical clustering and heatmaps of differentially expressed genes were generated in R using the heatmap.2 command in 'gplots' package, FPKM values were log2 transformed, and centered and scaled by rows for display purposes. Distance was calculated using the 'Maximum' method whereas clustering was performed using the 'Complete' method. Functional annotation clustering was performed using DAVID v6.7 and v6.8 (https://david.ncifcrf.gov/home.jsp; *Huang et al., 2009*). Gene set enrichment analysis was performed using GSEA v2 *Subramanian et al., 2005*), gene sets were obtained from the Broad Institute Molecular Signatures Database (http://software.broadinstitute.org/gsea/msigdb). MA plots were created in R using the ma.plot command in the 'affy' package, and MS vs RNAseq plots were created using the plot command.

## Tissue processing

Samples were fixed in 4% PFA. For cryosectioning, samples were incubated in the following series of solutions: 10% sucrose, 20% sucrose, 30% sucrose, 1:1 mixture of 30% sucrose and OCT (overnight), and OCT (1 hr). Samples were frozen in OCT. For microtome sectioning, samples were paraffin embedded in the histology core at Beth Israel Deaconess Medical Center.

## Immunohistochemistry

Cryosections were blocked and permeabilized (0.3% Triton-X-100 in PBS; 5% serum), incubated in primary antibodies overnight and secondary antibodies for 2 hr. Sections were counterstained with Hoechst 33342 and mounted using Fluoromount-G (SouthernBiotech, Birmingham, AL). The following primary antibodies were used: anti-5.8S rRNA (Y10b; Abcam, ab171119, 1:50), anti-BrdU (Biorad, MCA2060, 1:200), anti-cMYC (Abcam, ab32072, 1:100), anti-CTIP2 (Abcam, ab18465, 1:200), anti-Cux1 (Santa Cruz Biotechnology, sc13024, 1:200), anti-EIF3η (Santa Cruz Biotechnology, sc-137214, 1:100), anti-Fibrillarin (Abcam, ab4566, 1:250), anti-p4E-BP1 (Cell Signaling, 2855, 1:200), anti-Pax6 (Biolegend, 901301, 1:100), anti-pS6 (Cell Signaling, 5364, 1:200), anti-pVimentin (Enzo Bioscience,

ADI-KAM-CC249-E, 1:400), anti-Rpl11 (Santa Cruz Biotechnology, sc50363 1:50), anti-Rps12 (Proteintech, 16490–1-AP, 1:50), anti-Tuj1 (Biolegend, 801202, 1:100). Secondary antibodies were selected from the Alexa series (Invitrogen, 1:500). For BrdU, Fibrillarin, Pax6, cMyc, 5.8S rRNA, and EIF3η staining, antigen retrieval/denaturation was performed before the blocking step: A food steamer (Oster 5712) was filled with water and preheated until the chamber was approximately 100˚C, sections were immersed in boiling citric acid buffer (10 mM sodium citrate; 0.05% Tween 20; pH = 6) and placed in steamer for 20 min. Sections were cooled to room temperature. H&E staining was carried out according to standard procedures (*Shannon et al., 2018*).

## Immunoblotting

Tissues were homogenized in RIPA buffer supplemented with protease and phosphatase inhibitors. Protein concentration was determined by BCA assay (Thermo Scientific 23227). Samples were denatured in 2% SDS by heating at 95˚C for 5 min. Equal amounts of proteins were loaded and separated by electrophoresis in a 4–15% gradient polyacrylamide gel, transferred to nitrocellulose blot (250mA, 1.5 hr), blocked in 5% BSA or milk, incubated with primary antibodies overnight at 4˚C followed by HRP conjugated secondary antibodies (1:5000) for 1 hr, and visualized with ECL substrate. For phosphorylation analysis, the phospho-proteins were probed first, and then blots were stripped (Thermo Scientific 21059) and reprobed for total proteins. The following primary antibodies were used: anti-4E-BP1 (Cell Signaling, 9644, 1:1000), anti-ACTB (Cell Signaling, 12262, 1:2000), anti-cMYC (Abcam, ab32072, 1:2000), anti-GAPDH (Cell Signaling, 97166, 1:1000), anti-p4E-BP1 (Cell Signaling, 9459, 1:1000), anti-pS6 (Cell Signaling, 5364, 1:1000), anti-pS6K (Cell Signaling, 9234, 1:1000), anti-RPL10A (Novusbio, H00004736-M01, 1:500), anti-S6 (Cell Signaling, 2217, 1:1000), anti-S6K (Cell Signaling, 9202, 1:1000), anti-Vinculin (Cell Signaling, 13901, 1:1000).

## Fluorescent in situ hybridization

Cryosections were permeabilized in 0.5% Triton-X-100 for 20 min, incubated with probes (*Pilaz et al., 2016*) overnight at 37˚C, counterstained with Hoechst, and mounted using Fluoromount-G (SouthernBiotech). Probes: Quaser 570 coupled 5.8S pre-rRNA, Quaser 670 coupled 5.8S total rRNA.

## $^{35}$S-Methionine labeling

E8.5 and E10.5 forebrain neuroepithelium was dissected as described (*Chau et al., 2015*) and trypsinized. Cells were serum starved in methionine-free DMEM for 1 hr at 37˚C, then incubated with 51 µCi $^{35}$S-Methionine (Perkin Elmer NEG709A) at 37˚C for an additional hour. Cycloheximide (50 µg/ml) was added to stop translation. $^{35}$S-Methionine incorporation was measured using scintillation counter.

## Nucleolar volume quantification

Nucleolar volume was quantified according to published methods using Imaris (Bitplane; *Baker, 2013*; *Sanchez et al., 2016*; *Shannon et al., 2018*; *Silvera et al., 2010*). To ensure fair representation, randomly selected nucleoli were selected for quantification across the image field. When quantifying nucleolar volume embryonically, we specifically quantified cells close to the ventricular surface. Therefore, at E14.5 the quantified cells should represent radial glia in the ventricular zone. For relative nucleolar volume, each volume value was normalized to the average nucleolar volume of the controls in the corresponding litter. 5.8S rRNA signal area (nucleolar area) was quantified using FIJI (Image J).

## Neuroepithelium OPP quantification

OPP quantification was performed as described by *Liu et al. (2012)*. Pregnant dams received intraperitoneal OPP injections (50 mg/kg OPP; Life Technologies). One hour later, developing tissues were obtained and sectioned to a thickness of 7 µm using a cryostat. OPP signals were detected using the Click-iT plus OPP protein synthesis assay kits (Life Technologies) according the manufacturer's suggested procedures. Images were taken at 20X (Zeiss Axio Observer D1 inverted microscope) and fluorescence intensity was quantified using FIJI (ImageJ). For each sample, OPP intensity

from six independent regions of interest (185 µm²) along the ventricular surface was measured and averaged.

## MYC inhibitor injection

KJ-Pyr-9 (*Hart et al., 2014*) was dissolved in Tween 80:DMSO:5% dextrose (1:1:8) and injected at a dosage of 10 mg/kg into pregnant dams at E7.5. Samples were collected 24 hr later for analysis.

## BrdU cell proliferation assay

BrdU (50 mg/kg) was injected intraperitoneally into pregnant dams 2 hr prior to tissue collection. Brains were cryosectioned (7 µm thickness) and stained with BrdU and Pax6 antibodies. Images were acquired at 20X (Zeiss LSM 700 laser scanning confocal microscope). Cells were counted in a 100 µm wide column in the dorsal-lateral cortex. For each sample, 4–6 sections along the anterior/posterior axis of the forebrain were counted and averaged. The proliferation index was defined as the percentage of Pax6-positive cells that were also BrdU-positive.

## P0 cortical neuron counting

14 µm thick cryosections were stained with antibodies, and images were acquired at 20X (Zeiss Axio Observer D1 inverted microscope). Counting was performed using FIJI (Image J) on 100 µm wide columns in the dorsal-lateral cortex in the region just anterior to the hippocampus.

## P0 cortical thickness measurement

Measurements were performed on H and E-stained coronal sections. Thickness was defined as the length extending from the ventricular zone up to the pial surface in the dorsal-lateral cortex.

## Quantitative RT-PCR

RNA was isolated using Trizol extraction protocol or RecoverAll Total Nucleic Acid Isolation Kit (Ambion), and reverse-transcribed into cDNA. Gene expression was measured by Taqman qPCR (Life Technologies), using *Tbp* as an internal control.

## Transmission electron microscopy

All tissue processing, sectioning, and imaging was carried out at the Conventional Electron Microscopy Facility at Harvard Medical School. E8.5 and E10.5 tissues were fixed in 2.5% Glutaraldehyde/ 2% Paraformaldehyde in 0.1 M sodium cacodylate buffer (pH 7.4). They were then washed in 0.1M cacodylate buffer and postfixed with 1% Osmiumtetroxide (OsO4)/1.5% Potassiumferrocyanide (KFeCN6) for one hour, washed in water three times and incubated in 1% aqueous uranyl acetate for one hour. This was followed by two washes in water and subsequent dehydration in grades of alcohol (10 min each; 50%, 70%, 90%, 2 × 10 min 100%). Samples were then incubated in propyleneoxide for one hour and infiltrated overnight in a 1:1 mixture of propyleneoxide and TAAB Epon (Marivac Canada Inc. St. Laurent, Canada). The following day, the samples were embedded in TAAB Epon and polymerized at 60 degrees C for 48 hr. Ultrathin sections (about 80 nm) were cut on a Reichert Ultracut-S microtome, and picked up onto copper grids stained with lead citrate. Sections were examined in a JEOL 1200EX Transmission electron microscope or a TecnaiG[2] Spirit BioTWIN. Images were recorded with an AMT 2 k CCD camera.

Ribosomal quantification was performed using Imaris (Bitplane). For 20 images per individual (N = 3 at each age), ribosomal density was calculated within a 280.5 nm x 280.5 nm box in an inverted color image that contained only cytoplasm and ribosomes (no membrane bound organelles). Ribosomes were counted by the Imaris software using the 'spots' tool, with estimated diameter of 250px and with automatic background subtraction 'on', and quality above the automatic threshold. The number of ribosomes per field of view (FOV) was calculated by multiplying the above calculate density by the cytoplasmic area. The cytoplasmic area was calculated by creating a hand-drawn surface in Imaris around the free cytoplasmic space in the standard FOV (2692 nm x 1762.6 nm). The % FOV occupied by organelles was calculated by subtracting the free cytoplasmic area from the total area to arrive at the organelle-occupied area.

## Statistical analysis

Biological replicates (N) were defined as samples from distinct individuals analyzed either in the same experiment or within multiple experiments. Samples were pooled across multiple litters so as to reduce inter-litter variability. Statistical analyses were performed using Prism seven or R. Outliers were excluded using ROUT method (Q = 1%). Appropriate statistical tests were selected based on the distribution of data, homogeneity of variances, and sample size. F tests or Bartlett's tests were used to assess homogeneity of variances between data sets. Parametric tests (T test, ANOVA) were used only if data were normally distributed and variances were approximately equal. Otherwise, non-parametric alternatives were chosen. Data are presented as means ± standard errors of the mean (SEMs). Please refer to figure legends for statistical tests used and sample size. P values < 0.05 were considered significant (*$p \leq 0.05$, **$p \leq 0.01$, ***$p \leq 0.001$, ****$p \leq 0.0001$)

## Acknowledgements

We thank members of the Lehtinen and Fleming labs, C Harwell, A LaMantia, S Dymecki, A Lassar, and R Segal for helpful discussions, T Baudino for *Myc* knockout mice, D Silver for FISH probes, J Steen for RPS12 and RPL11 antibodies, M Baizabal, M Ericsson, A Malesz, and P Schmidt for experimental advice and assistance. We are grateful for the following support: NSF Graduate Research Fellowship (KFC), NIH T32 HL110852 (KFC and RMF), Pediatric Hydrocephalus Foundation, Simons Foundation SFARI Pilot Grant #402089, NIH R01 NS088566 (MKL), BCH IDDRC 1U54HD090255, and the New York Stem Cell Foundation. MK Lehtinen is a New York Stem Cell Foundation – Robertson Investigator.

## Additional information

### Funding

| Funder | Grant reference number | Author |
|---|---|---|
| National Science Foundation | Graduate Research Fellowship | Kevin F Chau |
| National Institutes of Health | R01 NS088566 | Maria K Lehtinen |
| Pediatric Hydrocephalus Foundation | Research grant | Maria K Lehtinen |
| Simons Foundation | SFARI Pilot Grant #402089 | Maria K Lehtinen |
| New York Stem Cell Foundation | Robertson Investigator | Maria K Lehtinen |
| National Institutes of Health | NIH T32 HL110852 | Kevin F Chau Ryann M Fame |

The funders had no role in study design, data collection and interpretation, or the decision to submit the work for publication.

### Author contributions

Kevin F Chau, Conceptualization, Resources, Data curation, Formal analysis, Supervision, Funding acquisition, Validation, Investigation, Visualization, Methodology, Writing—original draft, Project administration, Writing—review and editing; Morgan L Shannon, Data curation, Formal analysis, Validation, Investigation, Visualization, Methodology, Writing—review and editing; Ryann M Fame, Conceptualization, Data curation, Formal analysis, Supervision, Validation, Visualization, Methodology, Writing—review and editing; Erin Fonseca, Data curation, Formal analysis, Visualization; Hillary Mullan, Data curation, Validation, Visualization; Matthew B Johnson, Resources, Data curation, Software, Formal analysis, Supervision; Anoop K Sendamarai, Formal analysis, Supervision, Investigation, Writing—review and editing; Mark W Springel, Formal analysis, Validation, Investigation; Benoit Laurent, Data curation, Formal analysis, Supervision, Investigation; Maria K Lehtinen, Conceptualization, Supervision, Funding acquisition, Investigation, Visualization, Methodology, Writing—original draft, Writing—review and editing

## Author ORCIDs

Kevin F Chau (iD) http://orcid.org/0000-0002-7899-3247
Ryann M Fame (iD) http://orcid.org/0000-0002-8244-2624
Matthew B Johnson (iD) http://orcid.org/0000-0001-6909-5712
Maria K Lehtinen (iD) http://orcid.org/0000-0002-7243-2967

## Ethics

Animal experimentation: All animal experimentation was carried out under protocols approved by the IACUC of Boston Children's Hospital (protocol number 17-10-3547R).

## Decision letter and Author response

Decision letter https://doi.org/10.7554/eLife.36998.017
Author response https://doi.org/10.7554/eLife.36998.018

## Additional files

### Supplementary files

• Supplementary file 1. E8.5 vs E10.5 neuroepithelium RNA sequencing data: All genes (sheet 1), differentially expressed genes (DEG, sheet 2), DAVID functional annotation clustering (FAC, sheet 3 and 4), gene lists used for MA plot (sheet 5–10).
DOI: https://doi.org/10.7554/eLife.36998.011

• Supplementary file 2. WT vs MYC-OE apical progenitors RNA sequencing data: All genes (sheet 1), DEG (sheet 2), FAC of MYC-OE enriched genes (sheet 3), ribosomal protein genes used for MA plot in *Figure 5L* (Sheet 4), 53 genes that are enriched in both E8.5 and MYC-OE (Sheet 5).
DOI: https://doi.org/10.7554/eLife.36998.012

• Transparent reporting form
DOI: https://doi.org/10.7554/eLife.36998.013

### Data availability

Sequencing data have been deposited in GEO under accession number GSE100421.

The following dataset was generated:

| Author(s) | Year | Dataset title | Dataset URL | Database, license, and accessibility information |
|---|---|---|---|---|
| Chau KF, Johnson MB, Springel MW, Lehtinen MK | 2018 | Transcriptomic analysis of neuroepithelium and sorted neural progenitors in the murine cortex duirng early stages of development | https://www.ncbi.nlm.nih.gov/geo/query/acc.cgi?acc=GSE100421 | Publicly available at the NCBI Gene Expression Omnibus (accession no: GSE100421) |

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
