## [Decision Letter]

[Editors’ note: a previous version of this study was rejected after peer review, but the authors submitted for reconsideration. The first decision letter after peer review is shown below.]

Thank you for submitting your work entitled "Dynamic changes in protein biosynthesis during cell fate restriction of neural progenitors and in tumorigenesis" for consideration by *eLife*. Your article has been evaluated by a Senior Editor and three reviewers, one of whom is a member of our Board of Reviewing Editors.

Our decision has been reached after consultation between the reviewers. Based on these discussions and the individual reviews below, we regret to inform you that your work will not be considered further for publication in *eLife*.

The reviewers expressed enthusiasm for the conceptual advance and the timeliness of the studies. They acknowledge how these studies add to emerging work on the importance of translational regulation in the control of mammalian development and new concepts related to the control imposed on cell and tissue formation through specific ribosomal subunits. They also appreciate the variety of state of the art techniques and the initial transcriptional and proteomic experiments. However, in its current state, the paper is quite descriptive, somewhat correlative, the follow up analysis focusing on translation and the role of c-Myc in the process are lacking, and the manuscript is unfocused at the end.

The full reviews are attached below. A synopsis of the major critiques is outlined in the following 4 points.

1) The authors only directly examine whether protein synthesis is decreased post-neurulation (by the incorporation of OPP in Figure 2H) one time in the manuscript. The authors often use instead increased nucleolar volume (as a proxy) and EM analysis. The EM data need annotations and ideally immuno-gold to support the conclusion that punctae are actually ribosomes. It would be ideal if the authors could support their conclusion with standard molecular analysis of biogenesis – for example, are rRNA levels and/or processing different across development – and/or other direct measurements of protein synthesis and functional ribosomes (monosomes vs. polysomes).

2) Myc OE in apical progenitors is employed to support the idea that dynamic regulation of ribosome biogenesis and protein synthesis is needed in neural progenitors over development. However, these data are largely correlative and it remains unclear mechanistically whether alterations in ribosomes and protein synthesis explain the Myc OE phenotypes. Although Myc has been previously linked to ribosome biogenesis, this transcription factor has other functions, for example in associating with and impacting cell cycle regulators. Thus, the described alterations in brain size, Pax6+ S phase cells, and Cux1 number could be due to cell cycle alterations, potentially independent of ribosome regulation. Consistent with this possibility, Myc OE embryonic brains exhibit only a slight increase in ribosome biogenesis and no significant alterations in translation. The JQ1 experiments are problematic in interpretation of a specific effect on Myc, as JQ1 is a broad epigenetic (bromodomain) inhibitor and alterations in a number of epigenetic regulators can lead to neural tube defects. Can the authors build a case that better supports the title of the paper with additional ways of manipulating translational machinery directly to investigate their hypothesis that down-regulation of protein synthesis is needed for neurulation? Further experiments are needed to mechanistically link Myc modulation to ribosome biogenesis and/or to better link ribosome/translational regulation with cell fate restriction, in order to make a more impactful paper.

3) The Myc OE mouse model complements previous LOF findings that Myc controls brain size and progenitors (albeit using a human transgene). The authors conclude that Myc OE increases proliferation and in their title link dynamic changes in protein synthesis to "cell fate restriction." However, Myc OE phenotypes are only reported at later stages with analysis of only one neuronal marker, limiting the strength of the conclusions about cell fate restriction and proliferation. Are all neuronal layers affected? It would be important to demonstrate that Myc OE impacts proliferation and fate restriction at earlier relevant stages, E10.5-E12.5, when expression of Myc and ribosome/translation factors is reduced. *Foxg1-cre/myc* mice are also used which initiate c-myc expression early at e9.5 but there is little analysis of these embryos. In addition the data for Nestin-cre mice do not support the hypothesis that c-myc is driver of translation.

4) The reviewers would prefer that results related to Myc OE drives tumors in the choroid plexus is removed. These data diffuse the focus of the paper as the phenotypes and biology here are vastly different from early development. More attention on the early stages and requested experiments above would better strengthen the manuscript.

Despite the enthusiasm, the requirements suggested to flesh out the story could take considerable time to complete. If you feel it is possible to address these concerns, we encourage you to resubmit a more complete story for full review at a later date.

*Reviewer #1:*

This manuscript conceptually builds from the authors previous findings (Chau et al., 2015) that showed that the proteome of the amniotic fluid and early cerebrospinal fluid contains factors that promote placode and neural differentiation, respectively. The idea that arises in the current manuscript is that these factors derive from the developing neuroepithelium and the composition changes over time. Using RNA-seq, the authors profile the emerging forebrain in the mouse at E8.5 and E10.5. An added value of their analysis is the ability to compare the transcriptional datasets with their previous proteome analysis of the amniotic fluid and cerebrospinal fluid to make some inferences about signaling/reception and protein synthesis. The focus of the current manuscript is on the strong expression of ribosomal proteins at E8.5 and their downregulation by E10.5 as neurogenesis proceeds. It explores the role of MYC in this process using loss and gain of Myc expression. Together the manuscript presents novel and convincing data. Ultimately it is still somewhat correlative but the authors have done a nice job in highlighting the overall importance of MYC in the regulation of early development of the neuroepithelium and differentiation. These studies add to emerging work on the importance of translational regulation in the control of mammalian development and new concepts related to the exquisite control imposed on cell and tissue formation through specific ribosomal subunits.

The authors focus in on MYC as a possible upstream regulator. Most of the data in Figure 3 are solid but the JQ1 experiments are problematic in interpretation of a specific effect on Myc, as JQ1 is a broad epigenetic (bromodomain) inhibitor and alterations in a number of epigenetic regulators can lead to neural tube defects.

The results end on somewhat of a tangent and turns toward a cancer phenotype (which would be expected by MYC-OE) within the choroid plexus. Despite leading away from the main point, it is OK to retain this data as it shows a link between developmental and cancer pathways (again, not a surprise). However, since MYC overexpression is persistent through development, the following statement is incorrect or at best misleading: "The appropriate downregulation of Myc after neurulation is essential to prevent early missteps of development that can manifest as later-onset diseases of the nervous system."

*Reviewer #2:*

Here, Lehtinen and colleagues describe expression of ribosomal components and translation in neuroepithelial progenitors, which they associate with developmental restrictions in cell fate specification. Transcriptome analyses show down-regulation of these components between E8.5 and E10.5, which matches transcriptomes of the choroid plexus. This is parallel with changes in protein synthesis and proposed ribosome biogenesis. The authors suggest that reduced protein synthesis is required for differentiation to occur. They show that expression of Myc, a transcription factor implicated in ribosome biogenesis, correlates with these developmental profiles and demonstrate that in vivo Myc overexpression (Myc OE) perturbs brain development, increasing brain size and affecting neuronal markers. Myc OE in progenitors also drives tumors and alterations in translational machinery in the adult choroid plexus.

The main interesting concept put forth in this paper that ribosome biogenesis is developmentally regulated. Several additional concepts are intriguing: that CSF acquire translational components secreted from the neuroepithelium, and ribosome heterogeneity in progenitors (which has been suggested previously but poorly described). The data are of interest for those investigating neurulation and regulation of progenitor fate. However, in its current state, the paper is quite descriptive and somewhat unfocused, and the concepts described above could be better fleshed out.

1) Myc OE in apical progenitors is employed to support the idea that dynamic regulation of ribosome biogenesis and protein synthesis is needed in neural progenitors over development. However, these data are largely correlative and it remains unclear mechanistically whether alterations in ribosomes and protein synthesis explain the Myc OE phenotypes. Although Myc has been previously linked to ribosome biogenesis, this transcription factor has other functions, for example in associating with and impacting cell cycle regulators. Thus, the described alterations in brain size, Pax6+ S phase cells, and Cux1 number could be due to cell cycle alterations, potentially independent of ribosome regulation. Consistent with this possibility, Myc OE embryonic brains exhibit only a slight increase in ribosome biogenesis and no significant alterations in translation. Can the authors build a case that better supports the title of the paper with additional ways of manipulating translational machinery, for example? Further experiments are needed to mechanistically link Myc modulation to ribosome biogenesis and/or to better link ribosome/translational regulation with cell fate restriction, in order to make a more impactful paper.

2) The Myc OE mouse model complements previous LOF findings that Myc controls brain size and progenitors (albeit using a human transgene). The authors conclude that Myc OE increases proliferation and in their title link dynamic changes in protein synthesis to "cell fate restriction." However Myc OE phenotypes are only reported at later stages with analysis of only one neuronal marker, limiting strength of conclusions about cell fate restriction and proliferation. Are all neuronal layers affected? It would be important to demonstrate that Myc OE impacts proliferation and fate restriction at earlier relevant stages, E10.5-E12.5, when expression of Myc and ribosome/translation factors is reduced.

3) The last two figures of the paper demonstrate Myc OE drives tumors in the choroid plexus and modifies translation markers in these adult stages. While I appreciate this helps support the link between Myc and ribosome regulation, given the phenotypes and biology here are vastly different from early development, these data diffuse the focus of the paper.

4) The conclusion that E8.5 neuroepithelial progenitors have higher ribosome biogenesis than E10.5 is based upon increased nucleolar volume (as a proxy) and EM analysis. The EM data need annotations and ideally immuno-gold to support the conclusion that punctae are actually ribosomes. It would be ideal if the authors could support their conclusion with standard molecular analysis of biogenesis- for example, are rRNA levels and/or processing different across development?

*Reviewer #3:*

Chau et al. performed transcriptome analysis on forebrain e8.5 and e10.5 neuroepithelium and discovered that the protein biosynthetic machinery is down-regulated after neurulation. The authors then explore if c-myc regulates the expression of ribosomes and other translational machinery genes during neurulation. The work is extensive and the authors use a variety of state of the art techniques. Differential regulation of ribosomes and other translational machinery as a driver of neurulation is novel and could have a broad impact on our understanding of this important and complex morphogenetic process. While the initial transcriptional and proteomic experiments are performed well, the follow up analysis focusing on translation and the role of c-myc in the process are lacking.

The authors only directly examine whether protein synthesis is decreased post-neurulation (by the incorporation of OPP in Figure 2H) one time in the manuscript. The authors often use instead nucleolar data, which is interesting, but it is only correlative. A more direct examination of protein synthesis should also be performed on the c-myc knock-out and c-myc over-expression mice. Additional direct measurements of protein synthesis and functional ribosomes (monosomes vs. polysomes) are also needed in all instances to substantiate their findings.

The necessity of c-myc in coordinating the down-regulation of the protein synthesis machinery is also lacking. The authors use a conventional knock-out when a conditional knock-out might be more informative. The authors use JQ1 as an inhibitor of c-myc but concerns exist about the specificity of JQ1 for only c-myc. The authors could also use more standard and complementary systems that are likely to be more specific to test the necessity of c-myc. Unfortunately without these data, the author's conclusion (subsection “Critical timing of MYC expression during early forebrain development”, last paragraph) is not supported.

The over-expression studies are also lacking. The authors use an innovative c-myc line and two cre drivers (*Nestin* and *Foxg1*) but perform a detailed analysis for only the *Nestin-cre* driver. Unfortunately this does not seem to be the best mouse since c-myc is expression commences days after neurulation is complete. Instead the *Foxg1-cre*/myc mice seem better since c-myc expression initiates at e9.5 but the necessary studies are not performed. In addition the data for *Nestin-cre* mice do not support the hypothesis that c-myc is driver of translation and neurulation

C-myc also likely regulates many pathways, just not the protein synthesis machinery. Thus, the authors should also manipulate the translational machinery directly to investigate their hypothesis that down-regulation of protein synthesis is needed for neurulation.

Finally, a more minor concern, but still a possible significant problem, is that the difference in protein machinery expression could vary along the A-P and D-V/M-L axes. Since neurulation varies along the A-P axis and there are specific cell types along the M-L/D-V axis (hinge points, neural fold cells), differences could exist. Some data (Figure 2C) support this possibility. This caveat should be stated explicitly in the manuscript.

[Editors’ note: what now follows is the decision letter after the authors submitted for further consideration.]

Thank you for submitting your article "Downregulation of ribosome biogenesis and protein synthesis during early forebrain development" for consideration by *eLife*. Your article has been reviewed by three peer reviewers, and the evaluation has been overseen by Marianne Bronner as the Senior and Reviewing Editor. The following individual involved in review of your submission has agreed to reveal her identity: Lee Niswander (Reviewer #2).

The reviewers have discussed the reviews with one another and the Reviewing Editor has drafted this decision to help you prepare a revised submission. The reviewers agree that the manuscript is much improved but needs a few changes prior to acceptance.

Summary:

In this paper, Chau et al. delve into the important topic of developmental ribosome biogenesis. he authors demonstrate that ribosomal mRNA and proteins are higher at an early stage of forebrain development (E8.5) and decrease by the onset of neurogenesis (E10.5). Moreover, they make the case that c-MYC positively regulates ribosome biogenesis and that a consequence of c-MYC overexpression is an increase neuronal production and macrocephaly.

Essential revisions:

1) Please add to the Discussion some technical caveats of monitoring translation in vivo (some of which were mentioned in the response to reviewers) and as discussed in detail in the next paragraph. Also in the Discussion please stage whether there could be a function of altered ribosome biogenesis independent of modulating translation.

While authors have strong evidence that ribosome signature (and biogenesis) changes during development (as described in a manuscript that was not cited Kraushar et al., 2015, but also in the cited Kraushar et al., 2016), authors do not provide strong evidence that they disrupted dynamics of protein synthesis due to ribosome and/or MYC changes. For example, RPL11 is known to sit at the monosome but not to associate with polysome, while during stress conditions it binds to the DNA promoter of p53 oncogene (Slavov, Cell Reports, 2015; Mahata et al., Ocogene, 2015). In addition, critical experiments testing protein synthesis (e.g. OPP injection or methionine analysis), in current state do not exclude possibility that different transcription dynamics play role in the observed OPP and methionine changes. Here, authors may consider an additional line of evidence – they may want to discuss discrepancies between levels of mRNA expression and matching protein these mRNAs encode (e.g. using qRT-PCR and western).

2) Please remove "protein synthesis" from the title. Since OPP was injected intraperitoneally, the availability of OPP to progenitors may be different between E8.5 and 10.5 (e.g. similar to decrease of a number of proteins in AF and CSF by E10.5). Overall, there is no strong experimental evidence that protein synthesis is downregulated in early progenitors, while it may also be easily upregulated for selective number of other mRNAs whose protein expression was not tested. For example, total levels of ribosomal proteins can decrease but their association with polysomes may increase (Kraushar et al., 2015, 2016). The same concern goes for not seeing consistent, significant differences in OPP incorporation in Myc^-/-^ mice. Thus, it is critical to either remove "protein synthesis" from their title or perform additional experiments (e.g. experiments focusing on protein synthesis where transcription is transiently inhibited at its minimum).

3) Using immunohistochemistry, the authors show significant decrease for a number of ribosome biogenesis players and ribosomal proteins. Is there a ribosome protein that did not change or maybe even increased? These possible additional findings would be in line with EM pictures and EM analysis that clearly show large number of ribosomes in apical progenitors.

---

## [Author Response]

[Editors’ note: the author responses to the first round of peer review follow.]

The reviewers expressed enthusiasm for the conceptual advance and the timeliness of the studies. They acknowledge how these studies add to emerging work on the importance of translational regulation in the control of mammalian development and new concepts related to the control imposed on cell and tissue formation through specific ribosomal subunits. They also appreciate the variety of state of the art techniques and the initial transcriptional and proteomic experiments. However, in its current state, the paper is quite descriptive, somewhat correlative, the follow up analysis focusing on translation and the role of c-Myc in the process are lacking, and the manuscript is unfocused at the end.The full reviews are attached below. A synopsis of the major critiques is outlined in the following 4 points.1) The authors only directly examine whether protein synthesis is decreased post-neurulation (by the incorporation of OPP in Figure 2H) one time in the manuscript. The authors often use instead increased nucleolar volume (as a proxy) and EM analysis. The EM data need annotations and ideally immuno-gold to support the conclusion that punctae are actually ribosomes.

In the revised manuscript, we provide larger, annotated, representative EM images of from E8.5 and E10.5 neural progenitor cells (Figure 2K). We quantified the number of ribosomes and other organelles occupying the cytoplasmic space in neural precursors and found that the E8.5 cells contain more ribosomes per defined area of cytoplasm than E10.5 progenitors (Figure 2L), though ribosome density within the free cytoplasmic space, per se, was not different between these two ages (Figure 2M). In addition, more E10.5 cytoplasm than E8.5 cytoplasm was occupied by other organelles including endoplasmic reticulum, mitochondria, and Golgi (Figure 2N), indicating an overall shift in organelle landscape at this age.

We are confident that the punctae in our images are ribosomes, and based on morphological characteristics, are very unlikely to be any other structures. For example glycogen, which we often observe in our EM analyses of other cell types such as choroid plexus epithelial cells, is larger, more irregular in shape, aggregates more, and is typically more electron dense. We consulted with additional experts on ribosomes and EM in the HMS community including:

Maria Ericsson (manager, Harvard Medical School EM Facility, where we perform our EM analyses)

Mark Fleming, MD, DPhil (Boston Children’s Hospital; e.g. Nguyen et al., Science 2017)

Wei-Chung Lee, PhD (HMS/BCH Neurobiology; Lee et al., Nature 2016)

These experts agree that the punctae in our samples are ribosomes. In addition, there was a general consensus that because our samples are so packed with ribosomes, immuno-gold EM would be problematic to interpret since the overall signal would likely be too strong/saturated. Together with our revised representative images and quantification (see above), we hope the reviewers will agree that ribosome biogenesis is decreased in the E8.5 to E10.5 forebrain.

It would be ideal if the authors could support their conclusion with standard molecular analysis of biogenesis – for example, are rRNA levels and/or processing different across development – and/or other direct measurements of protein synthesis and functional ribosomes (monosomes vs. polysomes).

We reached out to Debby Silver (Duke University), to discuss this point and to get her advice on reagents that may work well in the developing forebrain at these early ages. The Silver lab kindly shared FISH probes against 5.8S pre-rRNA and 5.8S total rRNA (Pilaz et al.,), which reveal more rRNA signal in E8.5 than E10.5 progenitors. These new data are shown in new Figure 2F, 2G.

We purchased the Y10b antibody used by many labs in the field (e.g. Kondrashov et al., 2011) and found that it, too, showed stronger immunofluorescence signal in E8.5 progenitors than E10.5 progenitors. These new data are shown in new Figure 2H.

We quantified ^35^S-methionine incorporation in vitro in forebrain progenitors, and observed higher ^35^S-methionine incorporation at E8.5 compared to E10.5 (counts per million cells, presented as E8.5 fold change normalized to E10.5 forebrain: Expt. 1 = 2.0-fold; Expt. 2 = 1.4-fold; Expt. 3 = 1.1-fold), indicative of a higher protein synthesis rate in E8.5 forebrain progenitor cells. These new data are described in the last paragraph of the subsection “Decreased ribosome biogenesis and protein synthesis in E10.5 neuroepithelium”. We have also added a new author to our study, Anoop Sendamarai (Fleming lab, BCH/HMS), as we collaborated with a lab with specific expertise in performing this experiment and that had a radioactivity license.

We reached out to experts in the Harvard Medical School community regarding the technical requirements for analyzing monosomes and polysomes. From our pilot dissections and discussions with Drs. Dan Finley and Mark Fleming (e.g. Nguyen et al., Science 2017), we have concluded that these analyses are technically not feasible given the very limited amounts of E8.5 presumptive forebrain that can be micro-dissected across similarly staged embryos and pooled across multiple litters of mice. However, as we provide new data for the rest of the points raised above, we hope that the reviewers agree that our manuscript is strengthened by the inclusion of these new data and that polysome profiling is not necessary to demonstrate that ribosome biogenesis is downregulated from E8.5 to E10.5.

2) Myc OE in apical progenitors is employed to support the idea that dynamic regulation of ribosome biogenesis and protein synthesis is needed in neural progenitors over development. However, these data are largely correlative and it remains unclear mechanistically whether alterations in ribosomes and protein synthesis explain the Myc OE phenotypes. Although Myc has been previously linked to ribosome biogenesis, this transcription factor has other functions, for example in associating with and impacting cell cycle regulators. Thus, the described alterations in brain size, Pax6+ S phase cells, and Cux1 number could be due to cell cycle alterations, potentially independent of ribosome regulation. Consistent with this possibility, Myc OE embryonic brains exhibit only a slight increase in ribosome biogenesis and no significant alterations in translation.

We agree that MYC has many roles in the cells, and that the large brain phenotype observed is very likely due to a combination of MYC’s effects on multiple signaling pathways. For example, MYC overexpression also led to the upregulation of genes involved with cell proliferation. For example, we observed upregulation of *Igf2* and *Insm1* in MYC-expressing progenitors. IGF2 regulates proliferation of cortical progenitor cells (e.g. Lehtinen et al., 2011), and *Insm1* is a transcription factor with roles in promoting the delamination of apical progenitors cells, thereby accelerating cortical development by promoting the formation of basal progenitors (e.g. Farkas et al., 2008; Tavano et al., 2018). We clarify this point in the first paragraph of the subsection “Persistent MYC expression increases progenitor proliferation, leading to macrocephaly”.

In the revised manuscript, we have added new experiments quantifying ribosome biogenesis (via Fibrillarin staining) in *Foxg1:MYC* mice as well as wild type mice treated with MYC inhibitor or vehicle control. These new, gain- and loss-of-function data, shown in new Figure 5E, 5M, confirm MYC’s modest but consistent regulation of ribosome biogenesis at this early stage of forebrain development. We performed analyses of OPP incorporation in *Foxg1:MYC* and *Nestin:MYC* as well as *Myc^-/-^* mice. In these analyses, we did not see consistent, significant differences in OPP incorporation, demonstrating MYC-regulation of ribosome biogenesis does not drive protein synthesis in these cells and that other mechanisms exist to regulate these processes. We mention these data in the last paragraph of the subsection “MYC modulates ribosome biogenesis in developing forebrain”.

The JQ1 experiments are problematic in interpretation of a specific effect on Myc, as JQ1 is a broad epigenetic (bromodomain) inhibitor and alterations in a number of epigenetic regulators can lead to neural tube defects.

We removed the previous JQ1 data from the revised manuscript. We present instead new data using a more specific MYC inhibitor (KJ-Pyr-9;Hart et al., 2014)and found that intraperitoneal delivery to the pregnant dam reduced nucleolar volume quantified by Fibrillarin staining. These new data are presented in new Figure 5E.

Can the authors build a case that better supports the title of the paper with additional ways of manipulating translational machinery directly to investigate their hypothesis that down-regulation of protein synthesis is needed for neurulation?

We have rewritten the text to clarify that our goal is *not* to investigate the process of neurulation. We also expand our studies to include new analyses of the mTOR signaling pathway, which is also downregulated between E8.5 and E10.5. These new data are presented in new Figure 3. In parallel, we also tested the consequences of maternal rapamycin injection during this early stage of development. In our injections of n=5 litters, we did not observe neural tube defects, though cannot rule out that they may occur. We have revised the Discussion section in the manuscript and hope that our revised Discussion clarifies these important points to the reader.

Further experiments are needed to mechanistically link Myc modulation to ribosome biogenesis and/or to better link ribosome/translational regulation with cell fate restriction, in order to make a more impactful paper.

We performed additional experiments to demonstrate MYC’s contributions to ribosome biogenesis by using (1) Selective MYC inhibitor KJ-Pyr-9 (see above), and (2) quantifying ribosome biogenesis in *Foxg1-MYC* mice, which has larger nucleoli. We clarified in the text that changes in ribosome biogenesis and MYC expression take place concurrently as neural progenitor progress to more lineage-committed fates. In the longer term, we observe a large brain phenotype in the mice, which may be due to a combination of MYCs actions on downstream signaling pathways (e.g. ribosomes but also enhanced expression of factors with known roles in development including *Igf2* and *Insm1*; see above).

3) The Myc OE mouse model complements previous LOF findings that Myc controls brain size and progenitors (albeit using a human transgene). The authors conclude that Myc OE increases proliferation and in their title link dynamic changes in protein synthesis to "cell fate restriction." However, Myc OE phenotypes are only reported at later stages with analysis of only one neuronal marker, limiting the strength of the conclusions about cell fate restriction and proliferation. Are all neuronal layers affected?

As requested, we performed additional analyses on cell cycle at earlier ages (E11.5) in *Foxg1-MYC* mice, but did not see significant differences in proliferation, consistent with the modest increase in brain size at E14.5. We also analyzed the cell identities contributing to the cerebral cortex in *Nestin-MYC* mice at P0 and P8. We binned the cortex into equal segments, and quantified the numbers of cells expressing the following laminar markers: Cux1 and Ctip2. At P0, we observed significantly more Cux1+ upper-layer neurons in the cortex of MYC-OE (Figure 6K, p≤0.01) but no difference in the number of Ctip2+ lower layer neurons (p=0.82). At P8, we also observed more Cux1+ neurons in MYC-OE, although the difference did not reach statistical significance (p=0.22). We did not observe differences in numbers of Ctip2+ neurons at P8 (p=0.96). These data suggest that the combinatorial effects of MYC overexpression most readily influence the later stages of neurogenesis, when upper layer neurons of the cerebral cortex are formed. However, we cannot rule out more subtle effects of cortical development that are not evaluated by these types of assays, or at the ages examined in this study. Our new data are included in the last paragraph of the subsection “Persistent MYC expression increases progenitor proliferation, leading to macrocephaly”. In the revised manuscript, we have focused on ribosome biogenesis during early stages of development and have shifted focus away from cell fate restriction.

It would be important to demonstrate that Myc OE impacts proliferation and fate restriction at earlier relevant stages, E10.5-E12.5, when expression of Myc and ribosome/translation factors is reduced. Foxg1-cre/myc mice are also used which initiate c-myc expression early at e9.5 but there is little analysis of these embryos. In addition the data for Nestin-cre mice do not support the hypothesis that c-myc is driver of translation.

We carried out the requested quantification of cell proliferation in *Foxg1-MYC* mice at E11.5, but did not observe a significant difference in proliferation, consistent with the modest increase in brain size at E14.5. *Foxg1-MYC* mice are embryonic lethal beginning soon after E14.5, preventing further analyses at later ages. While the *Foxg1-MYC* mice did show increased ribosome biogenesis, similar to the *Nestin-MYC* mice, they did not show differences in OPP incorporation. We explain this point in the subsection “Persistent MYC expression increases progenitor proliferation, leading to macrocephaly”. These data demonstrate that regulation of ribosome biogenesis does not necessarily lead to increases in protein synthesis.

4) The reviewers would prefer that results related to Myc OE drives tumors in the choroid plexus is removed. These data diffuse the focus of the paper as the phenotypes and biology here are vastly different from early development. More attention on the early stages and requested experiments above would better strengthen the manuscript.

We removed the last 2 figures of the original manuscript on choroid plexus tumors and placed more emphasis on the early stages of forebrain development in order to strengthen the manuscript. These changes are also reflected in our revised title, “Downregulation of ribosome biogenesis and protein synthesis during early forebrain development.”

Reviewer #1:[…] The authors focus in on MYC as a possible upstream regulator. Most of the data in Figure 3 are solid but the JQ1 experiments are problematic in interpretation of a specific effect on Myc, as JQ1 is a broad epigenetic (bromodomain) inhibitor and alterations in a number of epigenetic regulators can lead to neural tube defects.

We removed the previous JQ1 data from the revised manuscript. We present instead new data using a more specific MYC inhibitor(KJ-Pyr-9; see above)and found that intraperitoneal delivery to the pregnant dam reduced nucleolar volume quantified by Fibrillarin staining. These new data are presented in Figure 5E in the revised manuscript.

The results end on somewhat of a tangent and turns toward a cancer phenotype (which would be expected by MYC-OE) within the choroid plexus. Despite leading away from the main point, it is OK to retain this data as it shows a link between developmental and cancer pathways (again, not a surprise). However, since MYC overexpression is persistent through development, the following statement is incorrect or at best misleading: "The appropriate downregulation of Myc after neurulation is essential to prevent early missteps of development that can manifest as later-onset diseases of the nervous system."

As described above in our response to Major critique #4, we have removed the choroid plexus tumor portion of the study from the revised manuscript.

Reviewer #2:[…] 1) Myc OE in apical progenitors is employed to support the idea that dynamic regulation of ribosome biogenesis and protein synthesis is needed in neural progenitors over development. However, these data are largely correlative and it remains unclear mechanistically whether alterations in ribosomes and protein synthesis explain the Myc OE phenotypes. Although Myc has been previously linked to ribosome biogenesis, this transcription factor has other functions, for example in associating with and impacting cell cycle regulators. Thus, the described alterations in brain size, Pax6+ S phase cells, and Cux1 number could be due to cell cycle alterations, potentially independent of ribosome regulation.

As mentioned above in response to Major critique #2, we agree that MYC has many roles in the cells, and that the large brain phenotypes observed is very likely due to a combination of MYC’s effects on multiple signaling pathways. For example, MYC overexpression also led to the upregulation of genes involved with cell proliferation. We observed upregulation of Igf2 and Insm1 in MYC-expressing progenitors. IGF2 regulates proliferation of cortical progenitor cells (e.g. Lehtinen et al., Neuron 2011), and *Insm1* is a transcription factor with roles in promoting the delamination of apical progenitors cells, thereby accelerating cortical development by promoting the formation of basal progenitors (e.g. Farkas et al., 2008; Tavano et al., 2018). We explain this point in the first paragraph of the subsection “Persistent MYC expression increases progenitor proliferation, leading to macrocephaly”.

Consistent with this possibility, Myc OE embryonic brains exhibit only a slight increase in ribosome biogenesis and no significant alterations in translation. Can the authors build a case that better supports the title of the paper with additional ways of manipulating translational machinery, for example? Further experiments are needed to mechanistically link Myc modulation to ribosome biogenesis and/or to better link ribosome/translational regulation with cell fate restriction, in order to make a more impactful paper.

In the revised manuscript, we have added new experiments quantifying ribosome biogenesis (via Fibrillarin staining) in *Foxg1-MYC* mice as well as wild type mice treated with MYC inhibitor or vehicle control. These new, gain- and loss-of-function data, show in new Figure 5E, 5F, 5M, 5N, confirm MYC’s consistent regulation of ribosome biogenesis at this early stage of forebrain development. We performed analyses of OPP incorporation in *Foxg1:MYC* and *Nestin:MYC* as well as *Myc^-/-^* mice. In these analyses, we did not see consistent, significant differences in OPP incorporation, demonstrating MYC-regulation of ribosome biogenesis does not drive protein synthesis in these cells and that other mechanisms exist to regulate and perhaps protect cells from these processes. We believe it is particularly important to present these surprising findings to the field. We mention these data in the last paragraph of the subsection “MYC modulates ribosome biogenesis in developing forebrain”. In light of all our new data and analyses, we have revised the title of the manuscript to better reflect our findings, “Downregulation of ribosome biogenesis and protein synthesis during early forebrain development.”

2) The Myc OE mouse model complements previous LOF findings that Myc controls brain size and progenitors (albeit using a human transgene). The authors conclude that Myc OE increases proliferation and in their title link dynamic changes in protein synthesis to "cell fate restriction." However Myc OE phenotypes are only reported at later stages with analysis of only one neuronal marker, limiting strength of conclusions about cell fate restriction and proliferation. Are all neuronal layers affected? It would be important to demonstrate that Myc OE impacts proliferation and fate restriction at earlier relevant stages, E10.5-E12.5, when expression of Myc and ribosome/translation factors is reduced.

We performed additional analyses on *Foxg1-MYC* mice at E10.5-E11.5. These data did not uncover increased proliferation. However, in the *Foxg1-MYC* mice, the brain phenotype is only beginning to emerge at E14.5. These data suggest that any shifts in cell proliferation are very subtle and as mentioned above, MYC overexpression acts likely by multiple mechanisms to increase brain size.

3) The last two figures of the paper demonstrate Myc OE drives tumors in the choroid plexus and modifies translation markers in these adult stages. While I appreciate this helps support the link between Myc and ribosome regulation, given the phenotypes and biology here are vastly different from early development, these data diffuse the focus of the paper.

As described above in our response to Major critique #4, we removed the choroid plexus tumor portion of the study from the revised manuscript.

4) The conclusion that E8.5 neuroepithelial progenitors have higher ribosome biogenesis than E10.5 is based upon increased nucleolar volume (as a proxy) and EM analysis. The EM data need annotations and ideally immuno-gold to support the conclusion that punctae are actually ribosomes.

As mentioned above in response to Major critique #1, in the revised manuscript, we provide larger, annotated, representative EM images of from E8.5 and E10.5 neural progenitor cells (Figure 2K). We quantified the number of ribosomes and other organelles occupying the cytoplasmic space in neural precursors and found that the E8.5 cells contain more ribosomes per defined area of cytoplasm than E10.5 progenitors (Figure 2L), though ribosome density within the free cytoplasmic space, per se, was not different between these two ages (Figure 2M). In addition, more E10.5 cytoplasm than E8.5 cytoplasm was occupied by other organelles including endoplasmic reticulum, mitochondria, and Golgi (Figure 2N), indicating an overall shift in organelle landscape at this age.

We are confident that the punctae in our images are ribosomes, and based on morphological characteristics, are very unlikely to be any other structures. For example glycogen, which we often observe in our EM analyses of other cell types such as choroid plexus epithelial cells, is larger, more irregular in shape, aggregates more, and is typically more electron dense. We consulted with additional experts on ribosomes and EM in the HMS community including:

Maria Ericsson (manager, Harvard Medical School EM Facility, where we perform our EM analyses)

Mark Fleming, MD, DPhil (Boston Children’s Hospital; e.g. Nguyen et al., Science 2017)

Wei-chun Lee, PhD (HMS/BCH Neurobiology; Lee et al., Nature 2016)

These experts agree that the punctae in our samples are ribosomes. In addition, there was a general consensus that because our samples are so packed with ribosomes, immuno-gold EM would be problematic to interpret since the overall signal would likely be too strong/saturated. Together with our revised representative images and quantification (see above), we hope the reviewers will agree that ribosome biogenesis is decreased in the E8.5 to E10.5 forebrain.

It would be ideal if the authors could support their conclusion with standard molecular analysis of biogenesis- for example, are rRNA levels and/or processing different across development?

As mentioned above in response to Major critique #1, we reached out to Debby Silver (Duke University), to discuss this point and to get her advice on reagents that may work well in the developing forebrain at these early ages. The Silver lab kindly shared FISH probes against 5.8S pre-rRNA and 5.8S total rRNA (Pilaz et al., 2016), which reveal more intense rRNA signal in E8.5 than E10.5 progenitors. These new data are shown in new Figure 2F, 2G.

We purchased the Y10b antibody used by many labs in the field (e.g. Kondrashov et al., 2011) and found that it, too, showed stronger immunofluorescence signal in E8.5 progenitors than E10.5 progenitors. These new data are shown in new Figure 2H.

We quantified ^35^S-methionine incorporation in vitro in forebrain progenitors, and observed higher ^35^S-methionine incorporation at E8.5 compared to E10.5 (counts per million cells, presented as E8.5 fold change normalized to E10.5 forebrain: Expt. 1 = 2.0-fold; Expt. 2 = 1.4-fold; Expt. 3 = 1.1-fold), indicative of a higher protein synthesis rate in E8.5 forebrain progenitor cells. These new data are described in the last paragraph of the subsection “Decreased ribosome biogenesis and protein synthesis in E10.5 neuroepithelium”.

We reached out to experts in the Harvard Medical School community regarding the technical requirements for analyzing monosomes and polysomes. From our pilot dissections and discussions with Drs. Dan Finley and Mark Fleming (e.g. Nguyen et al., Science 2017), we have concluded that these analyses are technically not feasible given the very limited amounts of E8.5 presumptive forebrain that can be micro-dissected across similarly staged embryos and pooled across multiple litters of mice. However, as we provide new data for the rest of the other points raised above, we hope that the reviewer agrees that our manuscript is strengthened by the inclusion of these new data and that polysome profiling is not necessary to demonstrate that ribosome biogenesis is downregulated from E8.5 to E10.5.

Reviewer #3:Chau et al. performed transcriptome analysis on forebrain e8.5 and e10.5 neuroepithelium and discovered that the protein biosynthetic machinery is down-regulated after neurulation. The authors then explore if c-myc regulates the expression of ribosomes and other translational machinery genes during neurulation. The work is extensive and the authors use a variety of state of the art techniques. Differential regulation of ribosomes and other translational machinery as a driver of neurulation is novel and could have a broad impact on our understanding of this important and complex morphogenetic process. While the initial transcriptional and proteomic experiments are performed well, the follow up analysis focusing on translation and the role of c-myc in the process are lacking.The authors only directly examine whether protein synthesis is decreased post-neurulation (by the incorporation of OPP in Figure 2H) one time in the manuscript. The authors often use instead nucleolar data, which is interesting, but it is only correlative. A more direct examination of protein synthesis should also be performed on the c-myc knock-out and c-myc over-expression mice. Additional direct measurements of protein synthesis and functional ribosomes (monosomes vs. polysomes) are also needed in all instances to substantiate their findings.

As mentioned above in response to Major critique #1, we quantified ^35^S-methionine incorporation in vitro in forebrain progenitors, and observed higher ^35^S-methionine incorporation at E8.5 compared to E10.5 (counts per million cells, presented as E8.5 fold change normalized to E10.5 forebrain: Expt. 1 = 2.0-fold; Expt. 2 = 1.4-fold; Expt. 3 = 1.1-fold), indicative of a higher protein synthesis rate in E8.5 forebrain progenitor cells. These new data are described in the last paragraph of the subsection “Decreased ribosome biogenesis and protein synthesis in E10.5 neuroepithelium”.

We have invested tremendous effort to perform the OPP analyses in the gain and loss-of-function models in lab. First, we have bred more *Myc*-deficient mice for this analysis at E8.5. However, these studies have been very challenging. The mice breed poorly, have very small litters, and as *Myc*-deficiency leads to small size, neural tube closure defects, and developmental delay, we have been faced with limited samples that are at similar developmental stages for our analyses. From the samples that we have collected, we have not observed reliable differences in OPP incorporation. Second, we have performed additional OPP quantification experiments in *Foxg1:MYC* mice. As we showed in the initial manuscript submission (with *Nestin:MYC* mice), MYC overexpression also did not result in clear upregulation of protein synthesis. We describe these observations in the last paragraph of the subsection “MYC modulates ribosome biogenesis in developing forebrain”. These observations differ from our hypothesis, which was formed based on findings in the cancer literature. Given our findings, we have focused the contributions of Myc expression in our study to be on ribosome biogenesis.

We also reached out to experts in the Harvard Medical School community regarding the technical requirements for analyzing monosomes and polysomes. From our pilot dissections and discussions with Drs. Dan Finley and Mark Fleming (e.g. Nguyen et al., Science 2017), we have concluded that these analyses are technically not feasible given the very limited amounts of E8.5 presumptive forebrain that can be micro-dissected across similarly staged embryos and pooled across multiple litters of mice. However, as we provide new data for the rest of the points raised above, we hope that the reviewers agree that our manuscript is improved and that polysome profiling is not necessary to demonstrate that ribosome biogenesis is downregulated from E8.5 to E10.5.

The necessity of c-myc in coordinating the down-regulation of the protein synthesis machinery is also lacking. The authors use a conventional knock-out when a conditional knock-out might be more informative. The authors use JQ1 as an inhibitor of c-myc but concerns exist about the specificity of JQ1 for only c-myc. The authors could also use more standard and complementary systems that are likely to be more specific to test the necessity of c-myc. Unfortunately without these data, the author's conclusion (subsection “Critical timing of MYC expression during early forebrain development”, last paragraph) is not supported.

As reviewer #1 raised concerns about the JQ1 MYC inhibitor due to its broader role as a bromodomain inhibitor, we have removed the previous JQ1 data from the revised manuscript. We present new data using KJ-Pyr-9, a more specific MYC inhibitorand found that intraperitoneal delivery to the pregnant dam reduced nucleolar volume quantified by Fibrillarin staining. These new data are presented in Figure 5E in the revised manuscript. These findings are in agreement with our data showing the *Myc*-deficient mice have smaller nucleoli as well (Figure 5F). While we do have the conditional *Myc* knockout mouse in lab, the field lacks a Cre driver line that would induce Cre recombination specifically in the neuroepithelium at the critical time point in development, just before neurulation. In the revised manuscript, we have removed the section regarding neural tube closure.

The over-expression studies are also lacking. The authors use an innovative c-myc line and two cre drivers (Nestin and Foxg1) but perform a detailed analysis for only the Nestin-cre driver. Unfortunately this does not seem to be the best mouse since c-myc is expression commences days after neurulation is complete. Instead the Foxg1-cre/myc mice seem better since c-myc expression initiates at e9.5 but the necessary studies are not performed. In addition the data for Nestin-cre mice do not support the hypothesis that c-myc is driver of translation and neurulation.

As requested, we have now performed additional analyses of nucleolar size and OPP incorporation in the *Foxg1:MYC* mice. We found that MYC expression in this line also resulted in increased nucleolar size (new Figure 5M), but as mentioned above, we did not observe differences in OPP incorporation (included in the last paragraph of the subsection “MYC modulates ribosome biogenesis in developing forebrain”). We agree with the reviewer that based on these findings, MYC does not appear to be a driver of translation in this system, although there may be other regulatory steps including accessibility to chromatin contributing to these results. We discuss this point in the Discussion section as well.

C-myc also likely regulates many pathways, just not the protein synthesis machinery. Thus, the authors should also manipulate the translational machinery directly to investigate their hypothesis that down-regulation of protein synthesis is needed for neurulation.

We agree that Myc has many roles and likely contributes to brain development in number of ways. We have rewritten the text to clarify that our goal is not to investigate the process of neurulation. We also expand our studies to include new analyses of the mTOR signaling pathway, which is also downregulated between E8.5 and E10.5. These new data are presented in new Figure 3. In parallel, we also tested the consequences of maternal rapamycin injection to inhibit the mTOR signaling pathway during this early stage of development. In our injections of n=5 litters, we did not observe neural tube defects, though cannot rule out that they may occur with different doses or other modifications to the protocol. We have revised the Discussion section in the manuscript and hope that our revised Discussion clarifies these important points to the reader.

Finally, a more minor concern, but still a possible significant problem, is that the difference in protein machinery expression could vary along the A-P and D-V/M-L axes. Since neurulation varies along the A-P axis and there are specific cell types along the M-L/D-V axis (hinge points, neural fold cells), differences could exist. Some data (Figure 2C) support this possibility. This caveat should be stated explicitly in the manuscript.

This is an interesting and important point. We have included this in the fifth paragraph of the Discussion section.

[Editors' note: the author responses to the re-review follow.]

Essential revisions:1) Please add to the Discussion some technical caveats of monitoring translation in vivo (some of which were mentioned in the response to reviewers) and as discussed in detail in the next paragraph. Also in the Discussion please stage whether there could be a function of altered ribosome biogenesis independent of modulating translation.While authors have strong evidence that ribosome signature (and biogenesis) changes during development (as described in a manuscript that was not cited Kraushar et al., 2015, but also in the cited Kraushar et al., 2016), authors do not provide strong evidence that they disrupted dynamics of protein synthesis due to ribosome and/or MYC changes. For example, RPL11 is known to sit at the monosome but not to associate with polysome, while during stress conditions it binds to the DNA promoter of p53 oncogene (Slavov, Cell Reports, 2015; Mahata et al., Ocogene, 2015). In addition, critical experiments testing protein synthesis (e.g. OPP injection or methionine analysis), in current state do not exclude possibility that different transcription dynamics play role in the observed OPP and methionine changes. Here, authors may consider an additional line of evidence – they may want to discuss discrepancies between levels of mRNA expression and matching protein these mRNAs encode (e.g. using qRT-PCR and western).

We have added a new paragraph to the Discussion section to address these concerns.

2) Please remove "protein synthesis" from the title. Since OPP was injected intraperitoneally, the availability of OPP to progenitors may be different between E8.5 and 10.5 (e.g. similar to decrease of a number of proteins in AF and CSF by E10.5). Overall, there is no strong experimental evidence that protein synthesis is downregulated in early progenitors, while it may also be easily upregulated for selective number of other mRNAs whose protein expression was not tested. For example, total levels of ribosomal proteins can decrease but their association with polysomes may increase (Kraushar et al., 2015, 2016). The same concern goes for not seeing consistent, significant differences in OPP incorporation in Myc^-/-^ mice. Thus, it is critical to either remove "protein synthesis" from their title or perform additional experiments (e.g. experiments focusing on protein synthesis where transcription is transiently inhibited at its minimum).

We have removed “protein synthesis” from the title.

3) Using immunohistochemistry, the authors show significant decrease for a number of ribosome biogenesis players and ribosomal proteins. Is there a ribosome protein that did not change or maybe even increased? These possible additional findings would be in line with EM pictures and EM analysis that clearly show large number of ribosomes in apical progenitors.

Based on our RNAseq data, ribosomal proteins that are not differentially expressed between E8.5 and E10.5 include:

IDE8.5 FPKME10.5 FPKMFold_change (log2)Sig*Rpl13a*
481.511718.4520.577324no*Rpl7a*
1405.231227.52-0.195062no*Rpsa*
368.641292.213-0.335194no*Rpl4*
1519.981168.83-0.378986no

Using immunoblotting, we found that RPL10A, which has higher RNA expression at E8.5 (*Rpl10a* FPKM: E8.5 = 1962.03; E10.5 = 1242.73), has comparable protein expression between E8.5 and E10.5. The difference between mRNA and protein levels suggests the presence of additional post-transcriptional mechanisms that regulate of expression of this gene. This new blot is now shown in new Figure 2M.

Despite the down-regulation of the majority of ribosomal proteins, we did not observe changes in ribosome density by EM. These observations could also be explained by the long half-life of ribosomes, which can be as long as 5 days. We have also addressed this in the Discussion section.